# Efficient Risk-Averse Reinforcement Learning

**Ido Greenberg**
Technion
gido@campus.technion.ac.il

**Yinlam Chow**
Google Research
yinlamchow@google.com

**Mohammad Ghavamzadeh**
Google Research
ghavamza@google.com

**Shie Mannor**
Technion, Nvidia Research
shie@ee.technion.ac.il

## Abstract

In risk-averse reinforcement learning (RL), the goal is to optimize some risk measure of the returns. A risk measure often focuses on the worst returns out of the agent's experience. As a result, standard methods for risk-averse RL often ignore high-return strategies. We prove that under certain conditions this inevitably leads to a local-optimum barrier, and propose a mechanism we call soft risk to bypass it. We also devise a novel cross entropy module for sampling, which (1) preserves risk aversion despite the soft risk; (2) independently improves sample efficiency. By separating the risk aversion of the sampler and the optimizer, we can *sample* episodes with poor conditions, yet *optimize* with respect to successful strategies. We combine these two concepts in CeSoR – Cross-entropy Soft-Risk optimization algorithm – which can be applied on top of *any* risk-averse policy gradient (PG) method. We demonstrate improved risk aversion in maze navigation, autonomous driving, and resource allocation benchmarks, including in scenarios where standard risk-averse PG completely fails. Our results and CeSoR implementation are available on Github. The stand-alone cross entropy module is available on PyPI.

## 1 Introduction

Risk-averse reinforcement learning (RL) is important for high-stake applications, such as driving, robotic surgery, and finance [Vittori et al., 2020]. In contrast to risk-neutral RL, it optimizes a risk measure of the return random variable, rather than its expectation. A popular risk measure is the Conditional Value at Risk (CVaR), defined as $\text{CVaR}_\alpha(R) = \mathbb{E}[R \mid R \leq q_\alpha(R)]$, where $q_\alpha(R) = \inf\{x \mid F_R(x) \geq \alpha\}$ is the $\alpha$-quantile of the random variable $R$ and $F_R$ is its CDF. Intuitively, CVaR measures the expected return below a specific quantile $\alpha$, also termed the risk level. CVaR optimization is widely researched in the RL community, e.g., using adjusted policy gradient approaches (CVaR-PG) [Tamar et al., 2015b, Hiraoka et al., 2019]. In addition, CVaR is a coherent risk measure, and its optimization is equivalent to a robust optimization problem [Chow et al., 2015].

Since risk-averse RL aims to avoid the hazardous parts of the environment (e.g., dangerous areas in navigation), CVaR-PG algorithms typically sample a batch of $N$ trajectories (episodes), and then optimize w.r.t. the mean of the $\alpha N$ trajectories with worst returns [Tamar et al., 2015b, Rajeswaran et al., 2017]. This approach suffers from two major drawbacks: (i) $1 - \alpha$ of the batch is wasted and excluded from the optimization (where often $0.01 \leq \alpha \leq 0.05$), leading to sample inefficiency; (ii) focusing on the worst episodes inherently overlooks good agent strategies corresponding to high returns – a phenomenon we refer to as the *blindness to success*.

**An illustrative example – the Guarded Maze**: Consider the Guarded Maze benchmark (visualized in Figure 1d). The goal is to reach the target zone (a constant location marked in green), resulting in

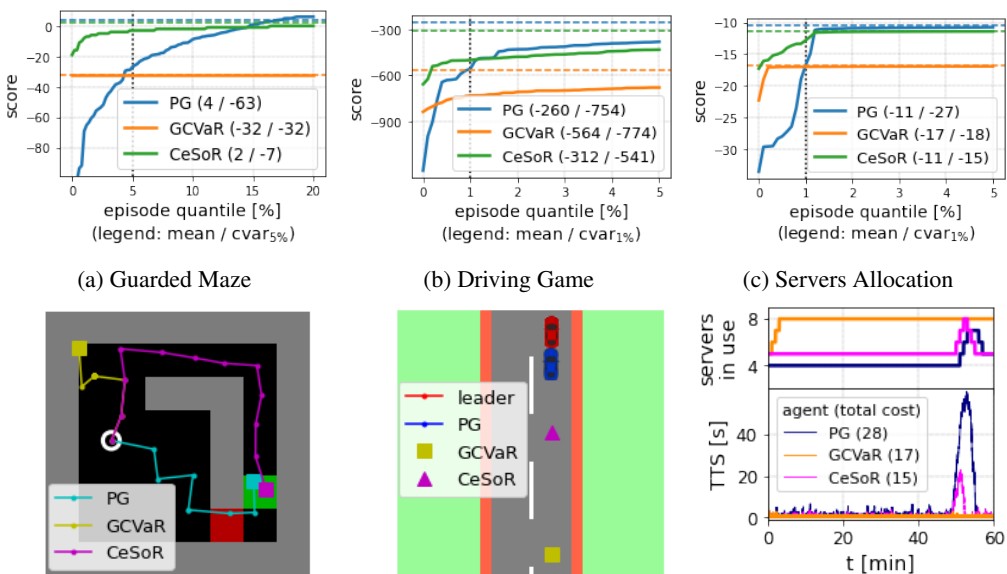

(a) Guarded Maze       (b) Driving Game       (c) Servers Allocation

(d) CeSoR learns to avoid the risk (red) and take the long path to the target (green), whereas GCVaR fails due to *blindness to success*.

(e) CeSoR maintains a safe margin from the leader, while PG has an accident and GCVaR maintains a too conservative distance.

(f) CeSoR handles the exceptional peak in user-requests without paying for as many servers as GCVaR, while PG fails to handle the peak.

Figure 1: Over 3 benchmarks, test results of 3 agents: risk-neutral PG, standard CVaR-PG (GCVaR, Tamar et al. [2015b]), and our CeSoR. Top: the lower quantiles of the returns distributions. Bottom: sample episodes.

a reward of 16 points. However, the guarded zone (in red) *may* be watched by a guard who demands a payment from any agent that passes by. Every episode, the probability that a guard is present is $\phi_1 = 20\%$, and the payment is exponentially-distributed with average $\phi_2 = 32$. That is, the cost of crossing the guarded zone in a certain episode is $C = C_1 \cdot C_2$, where $C_1 \sim Ber(\phi_1), C_2 \sim Exp(\phi_2)$ are independent and unknown to the agent. The agent starts at a random point at the lower half, and every time-step, observing its location, it selects an action: left, right, up or down, with an additive control noise. One point is deducted per step, up to 32 deductions.

In this maze, the shortest path maximizes the average return; yet, the longer path is CVaR-optimal, since sometimes short cuts make long delays [Tolkien, 1954]. However, the standard CVaR-PG optimizer (GCVaR in Figure 1) suffers from blindness to success: in a batch of $N$ random episodes, the worst $\alpha N$ returns (e.g., for $\alpha = 5\%$) usually correspond to either encountering a guard in the short path, or not reaching the goal at all. Hence, the desired long path is never even observed by the CVaR-PG optimizer, and cannot be learned.

Our key insight is that the variation in returns comes from both environment conditions (*epistemic* uncertainty) and agent actions (*aleatoric* uncertainty). We wish to focus on the *low quantiles w.r.t. the conditions* (e.g., a costly guard in the short path of the maze), yet to be exposed to the *high quantiles w.r.t. the strategies* (e.g., taking the long path in the maze). To that end, we devise two mechanisms: first, we use a soft risk-level scheduling method, which begins the training with risk neutrality $\alpha' = 1$, and gradually shifts the risk aversion to $\alpha' = \alpha$. Second, we present a novel dynamic-target version of the Cross Entropy method (CE or CEM) [de Boer et al., 2005], aiming to sample the worst parts of the environment. That is, the CEM samples trajectories with more challenging or riskier conditions, and the soft risk feeds a larger part of them ($\alpha' \geq \alpha$) to the CVaR-PG optimizer. Together, these constitute the Cross-entropy method for Soft-Risk optimization (*CeSoR*). CeSoR can be applied on top of any CVaR-PG method to learn any differentiable model (e.g., a neural network).

To apply the CEM, we assume to have certain control over the environment conditions. For example, in driving we may choose the roads for collecting training data, or in any simulation we may control the environment parameters (e.g., $\phi_1, \phi_2$ in the Guarded Maze). Note that (i) only the CE sampler (not the agent) is aware of the conditions; (ii) their underlying effect is unknown to the sampler and may vary with the agent throughout the training, hence the CEM needs to learn it adaptively.

**Contribution:** We present the following contribution for PG algorithms under risk-sensitive MDP problems (as defined in Section 2):

1. We analyze the phenomenon of *blindness to success* in the standard CVaR-PG, and show that it leads to a local-optimum barrier in certain environments (Section 3.1).

2. We analyze the potential increase in sample efficiency – if we could sample directly from the tail of the returns distribution (Section 3.2).

3. We introduce the CeSoR algorithm (Section 4), which modifies any CVaR-PG method with: (i) a soft risk mechanism preventing blindness to success; (ii) a novel dynamic-CE method that over-samples the riskier realizations of the environment, increasing sample efficiency.

4. We demonstrate the effectiveness of CeSoR in 3 risk-sensitive domains (Section 5), where it learns faster and achieves higher returns (both CVaR and mean) than the baseline CVaR-PG.

## 1.1 Related Work

Optimizing risk in RL is crucial to enforce safety in decision-making [García and Fernández, 2015, Paduraru et al., 2021]. It has been long studied through various risk criteria, e.g., mean-variance [Sato et al., 2001, Prashanth and Ghavamzadeh, 2013, 2016, Xie et al., 2018], entropic risk measure [Borkar and Meyn, 2002, Borkar and Jain, 2014, Fei et al., 2021] and distortion risk measures [Vijayan and Prashanth, 2021]. Tamar et al. [2015a] derived a PG method for general coherent risk measures, given their risk-envelope representation.

The CVaR risk measure specifically was studied using value iteration [Chow et al., 2015] and distributional RL [Dabney et al., 2018a, Tang et al., 2019, Bodnar et al., 2020] (also discussed in Appendix H). CVaR optimization was also shown equivalent to mean optimization under robustness [Chow et al., 2015], motivating robust-RL methods [Pinto et al., 2017, Godbout et al., 2021]. Yet, PG remains the most popular approach for CVaR optimization in RL [Tamar et al., 2015b, Rajeswaran et al., 2017, Hiraoka et al., 2019, Huang et al., 2021b], and can be flexibly applied to a variety of use-cases, e.g., mixed mean-CVaR criteria [Chow and Ghavamzadeh, 2014] and multi-agent problems [Qiu et al., 2021].

Optimizing the CVaR for risk levels $\alpha \ll 1$ poses a significant sample efficiency challenge, as only a small portion of the agent's experience is used to optimize its policy [Curi et al., 2020]. Keramati et al. [2020] used an exploration-based approach to address the sample efficiency. Pessimistic sampling for improved sample efficiency was suggested heuristically by Tamar et al. [2015b] using a dedicated value function, but no systematic method was suggested to direct the pessimism level. In this work, we use the CEM to control the sampled episodes around the desired risk level $\alpha$, and demonstrate CVaR optimization for as extreme levels as $\alpha = 1\%$. Note that unlike other CE-optimizers in RL [Mannor et al., 2003, Huang et al., 2021c], we use the CEM for *sampling*, to support a gradient-based optimizer.

## 2 Problem Formulation

Consider a Markov Decision Process (MDP) $(S, A, P, \gamma, P_0)$, corresponding to states, actions, state-transition and reward distribution, discount factor, and initial state distribution, respectively. For any policy parameter $\theta \in \mathbb{R}^n$, we denote by $\pi_\theta$ the parameterized policy that maps a state to a probability distribution over actions. Given a state-action-reward trajectory $\tau = \{(s_t, a_t, r_t)\}_{t=0}^{T}$, the trajectory total return is denoted by $R(\tau) = \sum_{t=0}^{T} \gamma^t r_t$. The expected return of a policy $\pi_\theta$ is defined as

$$J(\pi_\theta) = \mathbb{E}_{\tau \sim P^{\pi_\theta}} [R(\tau)], \tag{1}$$

where $P^{\pi_\theta}(\tau) = P_0(s_0) \prod_{t=0}^{T-1} P(s_{t+1}, r_t | s_t, a_t) \pi_\theta(a_t | s_t)$ is the probability distribution of $\tau$ induced by $\pi_\theta$. Under the risk-neutral objective, the PG method uses the gradient $\nabla_\theta J(\pi_\theta)$ to learn $\theta$, aiming to increase the probability of actions that lead to higher returns. In contrast, CVaR-PG methods aim to optimize the risk-averse $\text{CVaR}_\alpha$ objective (w.r.t. a given risk level $\alpha$):

$$J_\alpha(\pi_\theta) = \mathbb{E}_{\tau \sim P^{\pi_\theta}} [R(\tau) \,|\, R(\tau) \leq q_\alpha(R|\pi_\theta)], \tag{2}$$

where $q_\alpha(R|\pi_\theta)$ is the $\alpha$-quantile of the return random variable of policy $\pi_\theta$. Thus, CVaR-PG algorithms aim to improve the actions specifically for episodes whose returns are lower than $q_\alpha(R|\pi_\theta)$. Specifically, given a batch of $N$ trajectories $\{\tau_i\}_{i=1}^{N}$ whose empirical return quantile is $\hat{q}_\alpha = \hat{q}_\alpha(\{R(\tau_i)\}_{i=1}^{N})$, the CVaR gradient estimation is given by [Tamar et al., 2015b]:

$$\nabla_\theta \hat{J}_\alpha(\{\tau_i\}_{i=1}^{N}; \pi_\theta) = \frac{1}{\alpha N} \sum_{i=1}^{N} w_i \cdot \mathbf{1}_{R(\tau_i) \leq \hat{q}_\alpha} (R(\tau_i) - \hat{q}_\alpha) \sum_{t=0}^{T} \nabla_\theta \log \pi_\theta(a_{i,t}; s_{i,t}), \tag{3}$$

where $w_i = P^{\pi_\theta}(\tau_i)/f(\tau_i \,|\, \pi_\theta)$ is the importance sampling (IS) correction factor for $\tau_i$, if $\tau_i$ is sampled from a distribution $f \neq P^{\pi_\theta}$. Specifically, as discussed below, we modify the sample distribution using the cross entropy method over a context-MDP formulation of the environment.

**Context-MDP:** As mentioned above, we aim to focus on high-risk environment conditions. To discuss the notion of conditions, given a standard MDP, we extend its formulation to a Context-MDP (C-MDP) [Hallak et al., 2015], where the *context* is a set of variables that capture (part or all of) the randomness of the original MDP. We define the extension as $(S, A, \mathcal{C}, P_C, \gamma, P_0, D_{\phi_0})$, where $C \in \mathcal{C}$ is sampled from the context space $\mathcal{C}$ according to the distribution $D_{\phi_0}$ parameterized by $\phi_0$, and $P_C(\cdot) = P(\cdot|C)$ is the transition and reward distribution conditioned on $C$. In a C-MDP, a context-trajectory pair is sampled from the distribution $P^{\pi_\theta}_{\phi_0}(C, \tau) = D_{\phi_0}(C) P^{\pi_\theta}_C(\tau)$, where $P^{\pi_\theta}_C(\tau) = P_0(s_0) \prod_{t=0}^{T-1} P_C(s_{t+1}, r_t | s_t, a_t) \pi_\theta(a_t | s_t)$. The mean and CVaR$_\alpha$ objectives $J(\pi_\theta)$, $J_\alpha(\pi_\theta)$ in Equations (1) and (2) are naturally generalized to C-MDP using the distribution $P^{\pi_\theta}_{\phi_0}(C, \tau)$.

Once we extend an MDP into a C-MDP, we can learn how to modify the context-distribution parameter $\phi$ to sample high-risk contexts and trajectories, focusing the training on high-risk parts of the environment and thus improving sample efficiency. For this, we assume that certain aspects of the training environment (represented by $C$) can be controlled. This assumption indeed holds in many practical applications – in both simulated and physical environments. For example, consider a data collection procedure for a self-driving agent training, which by default samples all driving hours uniformly: $C \sim U([0, 24))$. As the hour may affect traffic and driving patterns, a risk-averse driver would prefer to sample more experience in high-risk hours. To that end, we could re-parameterize the uniform distribution as, say, $Beta(\phi)$ with $\phi_0 = (1, 1)$ (note that $Beta(1, 1)$ is the uniform distribution), learn the high-risk hours, and modify $\phi$ to over-sample them. As another example, in the Guarded Maze described above, we can control the parameters $\phi_1, \phi_2$ of the simulation.

# 3 Limitations of CVaR-PG

Consider the standard CVaR-PG algorithm, which relies on Equation (3) to apply PG for maximization of $J_\alpha(\pi_\theta)$ of (2). In this section, we analyze two major limitations of this algorithm. Section 3.1 analyzes the *blindness to success* phenomenon, which may bring CVaR-PG learning to a local-optimum deadlock. This will motivate the soft-risk scheduling in Section 4. Section 3.2 analyzes the potential increase in sample efficiency when the environmental context is sampled in correspondence to the tail of the returns distribution. This will motivate the cross-entropy sampler in Section 4.

While the analysis focuses on CVaR-PG methods, Appendix H discusses Distributional RL algorithms for CVaR optimization, and demonstrates that similar limitations apply to these methods as well.

## 3.1 Blindness to Success

We formally analyze how the *blindness to success* phenomenon can bring the policy learning to a local-optimum deadlock by ignoring successful agent strategies.

Recall the $\alpha$-quantile of a return distribution $q^\pi_\alpha = \min\{r \,|\, F_{R(a)|\pi}(r) \geq \alpha\}$. We first introduce the notion of a *tail barrier*, corresponding to a returns-distribution tail with a constant value.

**Definition 1** (Tail barrier). Let $\alpha \in (0, 1]$. A policy $\pi$ has an $\alpha$-tail barrier if $\forall \alpha' \in [0, \alpha] : q^\pi_{\alpha'} = q^\pi_\alpha$.

Note that in any environment with a discrete rewards distribution, a policy is prone to having a tail barrier for some $\alpha > 0$. In existing CVaR-PG analysis [Tamar et al., 2015b], such barriers are often overlooked by assuming continuous rewards. For the Guarded Maze, Figure 13c in the appendix demonstrates how a standard CVaR-PG exhibits a 0.9-tail barrier, since as many as 90% of the trajectories reach neither the target nor the guard, and thus have identical low returns.

A tail barrier has a destructive effect on CVaR-PG. Consider a CVaR$_\alpha$ objective, and a policy $\pi$ with a $\beta$-tail barrier where $\beta > \alpha$. Intuitively, any infinitesimal change of $\pi$ cannot affect the CVaR return, since the returns infinitesimally-above $q^\pi_\alpha$ are identical to those below $q^\pi_\alpha$. That is, any tail barrier wider than $\alpha$ brings the CVaR-PG to a stationary point of type plateau. More formally, consider $\nabla_\theta \hat{J}_\alpha$ of Equation (3) with a $\beta$-tail barrier $\beta > \alpha$: any trajectory has either $\mathbf{1}_{R(\tau_i) \leq q^\pi_\alpha} = 0$ (if its return is above $q^\pi_\alpha$) or $R(\tau_i) - q^\pi_\alpha = 0$ (otherwise), hence the whole gradient vanishes. Such a loss plateau was also observed in a specific MDP in Section 5.1 of Huang et al. [2021a].

In practice, a discrepancy between $q_\alpha^\pi$ and its estimate $\hat{q}_\alpha(\{R(\tau_i)\})$ (used in Equation 3) may prevent the gradient from completely vanishing, if $q_\alpha^\pi = q_\beta^\pi < \hat{q}_\alpha(\{R(\tau_i)\})$. Otherwise, if $\hat{q}_\alpha(\{R(\tau_i)\}) \leq q_\beta^\pi$ in every subsequent iteration, the gradient remains zero, the policy cannot learn any further, and any trajectory returns beyond $q_\alpha^\pi$ will never be even propagated to the optimizer. We refer to this phenomenon as *blindness to success*.

**Definition 2** (Blindness to success). Let a risk level $\alpha \in (0,1)$ and a CVaR-PG training step $m_0 \geq 1$, and let $\beta \in (\alpha, 1)$. Denote by $\mathcal{T}, \Pi$ the spaces of trajectories and policies, respectively, and by $\{\tau_{m,i}\}_{i=1}^N \sim P^{\pi_m}$ the random trajectories in step $m \geq m_0$. We denote by $\mathcal{B}_{\alpha,\beta}^{m_0,n}$ the event of blindness to success in the subsequent $n$ steps (and the complementary event by $\neg\mathcal{B}_{\alpha,\beta}^{m_0,n}$):

$$\mathcal{B}_{\alpha,\beta}^{m_0,n} = \left\{ \left\{ \left( \{\tau_{m,i}\}_{i=1}^N, \pi_m \right) \right\}_{m_0 \leq m < m_0+n} \in (\mathcal{T}^N \times \Pi)^n \;\middle|\; \forall m: \; \hat{q}_\alpha(\{R(\tau_{m,i})\}) \leq q_\beta^{\pi_{m_0}} \right\}.$$

Note that Definition 2 uses $q_\beta^{\pi_{m_0}}$ (corresponding to step $m_0$) to bound the returns in training steps $m > m_0$, thus indeed represents training stagnation. Theorem 1 shows that given a $\beta$-tail barrier with $\beta > \alpha$, the probability that CVaR-PG avoids the blindness to success decreases exponentially with $\beta - \alpha$. For example, for $n = 10^6$, $\alpha = 0.05$, $\beta = 0.25$, and $N = 400$, we have $\mathbb{P}(\neg\mathcal{B}_{\alpha,\beta}^{m_0,n}) < 10^{-7}$.

**Theorem 1.** Under Definition 2's conditions, $\mathbb{P}\left( \neg\mathcal{B}_{\alpha,\beta}^{m_0,n} \;\middle|\; \pi_{m_0} \text{ has } \beta\text{-tail barrier} \right) \leq ne^{-2N(\beta-\alpha)^2}$.

*Proof sketch (see the full proof in Appendix A).* In every step $m$, we have $q_\beta^{\pi_{m_0}} < \hat{q}_\alpha(\{R(\tau_{m,i})\})$ only if at least $1 - \alpha$ of the returns are higher than $q_\beta^{\pi_{m_0}}$. We bound the probability of this event using the Hoeffding inequality (Lemma 1). In the complementary event the gradient is 0 (due to the barrier), thus the policy does not change, and the argument can be applied inductively to the next step. $\quad\square$

## 3.2 Variance Reduction and Sample Efficiency

As discussed in Section 2, an MDP can be often re-parameterized as a C-MDP. In terms of the C-MDP, CVaR-PG samples $N$ context-trajectory pairs from the distribution $P_{\phi_0}^{\pi_\theta}(C, \tau)$, and calculates the policy gradients with respect to the $\alpha N$ trajectories with the lowest returns. That is, CVaR-PG aims to follow the policy gradients corresponding to the tail distribution defined by

$$P_{\phi_0,\alpha}^{\pi_\theta}(C, \tau) = \alpha^{-1} \mathbf{1}_{R(\tau) \leq q_\alpha(R|\pi_\theta)} P_{\phi_0}^{\pi_\theta}(C, \tau) \tag{4}$$

Notice that by considering only $\alpha$ of the trajectories, CVaR-PG essentially suffers from $\alpha^{-1}$-reduction in sample efficiency in comparison to risk-neutral PG.

Proposition 1 shows that if we could sample trajectories directly from $P_{\phi_0,\alpha}^{\pi_\theta}$, we would reduce the variance of the policy gradient estimate (and thus increase the sample efficiency) back by a factor of $\alpha^{-1}$. This will motivate the CEM in Section 4, which will aim to modify $\phi$ such that $P_\phi^{\pi_\theta} \approx P_{\phi_0,\alpha}^{\pi_\theta}$.

**Proposition 1** (Variance reduction). If the quantile estimation error is negligible ($\hat{q}_\alpha = q_\alpha(R|\pi_\theta)$ in Equation (3)), then

$$\mathrm{Var}_{\tau_i \sim P_{\phi_0,\alpha}^{\pi_\theta}}\left(\nabla_\theta \hat{J}_\alpha(\{\tau_i\}_{i=1}^N; \pi_\theta)\right) \leq \alpha \cdot \mathrm{Var}_{\tau_i \sim P_{\phi_0}^{\pi_\theta}}\left(\nabla_\theta \hat{J}_\alpha(\{\tau_i\}_{i=1}^N; \pi_\theta)\right).$$

*Proof sketch (see the full proof in Appendix B).* Since the left term corresponds to the sample distribution $P_{\phi_0,\alpha}^{\pi_\theta}$, the corresponding IS weights are $w \equiv \alpha$ w.p. 1. When applying IS analysis to the expected value, $w$ cancels out the distributional shift (as in Equation 5), resulting in the same expected gradient estimate. When applying the same analysis to the variance, we begin with the square weight $w^2$, thus a $w = \alpha$ factor still remains after the distributional shift compensation. $\quad\square$

The variance reduction can be connected to sample efficiency through the convergence rate as follows. According to Theorem 5.5 in Xu et al. [2020], denoting the initial parameters by $\theta_0$, the convergence of any CVaR-PG algorithm can be written as $\mathbb{E}[\|\nabla_\theta J_\alpha(\pi_\theta)\|^2] \leq \mathcal{O}(\frac{J_\alpha(\theta) - J_\alpha(\theta_0)}{M}) + \mathcal{O}(\frac{\mathrm{Var}(\nabla_\theta \hat{J}_\alpha(\{\tau_i\}_{i=1}^N; \pi_\theta))}{\alpha N})$. Clearly, variance reduction of $\alpha$-factor linearly improves the second term. In particular, it cancels out the denominator's $\alpha$-factor attributed to tail sub-sampling, and brings the sample efficiency back to the level of the risk-neutral PG.

# 4 The Cross-entropy Soft-Risk Algorithm

Algorithm 1 presents our Cross-entropy Soft-Risk algorithm (***CeSoR***), which uses a PG approach to maximize $J_\alpha(\pi_\theta)$ in (2). CeSoR adds two components on top of CVaR-PG: *soft-risk scheduling* to address the blindness to success analyzed in Section 3.1, and *CE sampling* to address the sample efficiency analyzed in Section 3.2.

---

**Algorithm 1: CeSoR**

---

1   **Input**: risk level $\alpha$; context distribution $D_\phi$; original context parameter $\phi_0$; training steps $M$; trajectories sampled per batch $N$, where $\nu$ fraction of them is from the original $D_{\phi_0}$; smoothed CE quantile $\beta$; risk-level scheduling factor $\rho$

2   **Initialize:**   policy $\pi_\theta$,   $\phi \leftarrow \phi_0$,

3   $N_o \leftarrow \lfloor \nu N \rfloor$,   $N_s \leftarrow \lceil (1-\nu)N \rceil$

4   **for** $m$ *in* $1:M$ **do**

     // Sample contexts

5      Sample $\{C_{o,i}\}_{i=1}^{N_o} \sim D_{\phi_0}$,   $\{C_{\phi,i}\}_{i=1}^{N_s} \sim D_\phi$

6      $C \leftarrow (C_{o,1}, \ldots, C_{o,N_o}, C_{\phi,1}, \ldots, C_{\phi,N_s})$

7      $w_{o,i} \leftarrow 1, \ \forall i \in \{1, \ldots, N_o\}$

8      $w_{\phi,i} \leftarrow \frac{D_{\phi_0}(C_{\phi,i})}{D_\phi(C_{\phi,i})}, \ \forall i \in \{1, \ldots, N_s\}$

9      $w \leftarrow (w_{o,1}, \ldots, w_{o,N_o}, w_{\phi,1}, \ldots, w_{\phi,N_s})$

     // Sample trajectories

10     $\{\tau_{C_{o,i}}\}, \{\tau_{C_{\phi,i}}\} \leftarrow \text{run\_episodes}(\pi_\theta, C)$

     // Update CE sampler

11     $q \leftarrow \max(\hat{q}_\alpha(\{R(\tau_{C_{o,i}})\}), \hat{q}_\beta(\{R(\tau_{C_{\cdot,i}})\}))$

12     $\phi \leftarrow \arg\max_{\phi'} \sum_{i \leq N} w_i \, \mathbf{1}_{R(\tau_{C_i}) \leq q} \log D_{\phi'}(C_i)$

     // PG step (e.g., Eq. 6)

13     $\alpha' \leftarrow \max(\alpha, 1 - (1-\alpha) \cdot m/(\rho \cdot M))$

14     $q' \leftarrow \hat{q}_{\alpha'}(\{R(\tau_{C_{o,i}})\})$

15     $\theta \leftarrow \text{CVaR\_PG}(\pi_\theta, (\{\tau_{C_{o,i}}\}, \{\tau_{C_{\phi,i}}\}), w, q')$

---

**Soft-risk scheduler**: We set the policy optimizer (Line 15 in Algorithm 1) to use a soft risk level $\alpha'$ that gradually decreases from 1 to $\alpha$ (Line 13 and Figure 2). This is motivated by the blindness to success analyzed in Section 3.1: by modifying the risk level to $\alpha' > \alpha$, and specifically $\alpha' \approx 1$ at the beginning of training, we guarantee that there cannot be a wider tail barrier $\beta > \alpha'$. Thus, CeSoR can feed the optimizer with trajectories whose returns $q_\beta^\pi < R \leq q_{\alpha'}^\pi$ are higher than any constant tail; and since the fed returns are not constant, they do not eliminate the gradient. In this sense, CeSoR looks beyond local optimization-plateaus to prevent the blindness to success.

The scheduling defined in Line 13 and Figure 2 is heuristic. As demonstrated in Section 5, once we understand the limitation of blindness to success, this simple heuristic is sufficient to bypass the blindness. An adaptive $\alpha'$ scheduling that maximizes blindness prevention probability would require tighter concentration inequalities [Boucheron et al., 2013], and is left for future work.

**Cross Entropy Method (CEM)**: The CEM [de Boer et al., 2005] is a general approach to rare-event sampling and optimization, which we use to sample high-risk contexts and trajectories. First, we review the standard CEM in terms adjusted to our setting and notations (for a more general presentation, see Algorithm 2 in the appendix). Then, we discuss the limitations of the standard CEM in the RL settings, and present our dynamic, regularized version of the CEM.

Motivated by the sample efficiency analysis of Section 3.2, we wish to align the agent's experience with the $\alpha$ worst-case returns – by sampling contexts whose corresponding trajectory-returns are likely to be below $q_\alpha(R|\pi_\theta)$. That is, we wish to sample context-trajectory pairs from $P_{\phi_0,\alpha}^{\pi_\theta}$ of (4). To that end, the CEM searches for a value of $\phi$ for which $P_\phi^{\pi_\theta}$ is similar to $P_{\phi_0,\alpha}^{\pi_\theta}$. More precisely, it looks for $\phi^*$ that minimizes the KL-divergence (i.e., cross-entropy) between the two:

$$
\begin{aligned}
\phi^* &\in \arg\min_{\phi'} \ D_{KL}\big(P_{\phi_0,\alpha}^{\pi_\theta}(C,\tau) \,\|\, P_{\phi'}^{\pi_\theta}(C,\tau)\big) \\
&= \arg\max_{\phi'} \ \mathbb{E}_{(C,\tau)\sim P_{\phi_0}^{\pi_\theta}} \big[ \alpha^{-1} \mathbf{1}_{R(\tau) \leq q_\alpha(R|\pi_\theta)} \log D_{\phi'}(C) \big] \\
&= \arg\max_{\phi'} \ \mathbb{E}_{(C,\tau)\sim P_\phi^{\pi_\theta}} \big[ \alpha^{-1} w(C,\tau) \, \mathbf{1}_{R(\tau) \leq q_\alpha(R|\pi_\theta)} \log D_{\phi'}(C) \big],
\end{aligned}
\tag{5}
$$

where $P_{\phi'}^{\pi_\theta}(C,\tau) = D_{\phi'}(C)P_C^{\pi_\theta}(\tau)$ (Section 2), and $w(C,\tau) = \frac{P_{\phi_0}^{\pi_\theta}(C,\tau)}{P_\phi^{\pi_\theta}(C,\tau)} = \frac{D_{\phi_0}(C)}{D_\phi(C)}$ is the IS weight corresponding to the sample distribution $(C,\tau) \sim P_\phi^{\pi_\theta}$. The optimization problem in Equation (5) often reduces to a simple closed-form calculation: if $D_\phi$ is a Gaussian, for example, $\phi^*$ reduces to the weighted expectation and variance of $\{C \mid R(\tau) \leq q_\alpha\}_{C,\tau\sim P_\phi^{\pi_\theta}}$ with the IS weights $w(C,\tau)$.

Equation (5) may produce noisy results when estimated from data $\{(C_i, \tau_i)\}_{i=1}^N$, unless $N \gg \alpha^{-1}$, since only $\alpha N$ trajectory-samples satisfy $R(\tau) \leq q_\alpha$ and are used in the estimation. To address this,

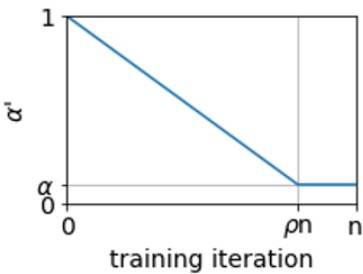

Figure 2: The soft-risk scheduling (Algorithm 1, Line 13). The linear phase $\alpha' > \alpha$ prevents the blindness to success (Section 3.1), while the CEM still preserves risk aversion. The final constant phase $\alpha' = \alpha$ provides a stationary objective and allows CeSoR to converge (Appendix C).

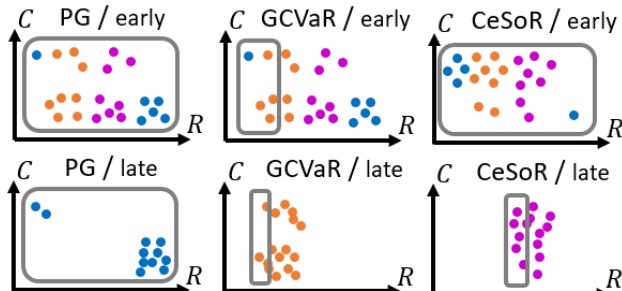

Figure 3: An illustration of training batches. Each point represents an episode with return $R$ and context $C$. Points of the same color correspond to "similar" agent actions that induce similar policy gradients. Mean-PG averages over the *whole* batch and learns the blue strategy. CVaR-PG considers the *left* part (low returns) and learns the orange strategy. CeSoR over-samples the *upper* part (high-risk contexts), and only later decreases $\alpha'$ to explicitly focus on low returns, thus learning the purple strategy. The illustrated episodes are analogous to the strategies in Figures 1d,4.

the CEM reaches the $\alpha$-tail gradually over iterations. Every iteration, it samples a batch of contexts $\{C_i\}_{i=1}^N$ from the current distribution $D_\phi$, and then solves Equation (5) with respect to a *higher* quantile $q \geq q_\alpha$. More specifically, denote by $\hat{q}_\alpha^\phi$ the estimated $\alpha$-quantile of $\{R(\tau)\}_{C,\tau \sim P_\phi^{\pi_\theta}}$; then, we set $q = \max(\hat{q}_\alpha^{\phi_0}, \hat{q}_\beta^\phi)$ with a hyperparameter $\beta > \alpha$ (often $\beta = 0.2$). Since the data is drawn from $P_\phi^{\pi_\theta}$, this guarantees at least $\beta N$ samples per update step. The quantile $\hat{q}_\alpha^{\phi_0}$ corresponds to the $\alpha$-tail of the original context-distribution, and can be viewed as a stopping condition: once $\hat{q}_\alpha^{\phi_0} > \hat{q}_\beta^\phi$, many of our samples are already in the tail, and $\beta$ is no longer needed to smooth the update of $\phi$.

**Dynamic-target CEM**: The standard CEM assumes to search for the tail of a *constant* distribution. In our setting, however, we look for the tail of the distribution of the returns $R(\tau)$, where $C, \tau \sim P_{\phi_0}^{\pi_\theta}$ depend on $\pi_\theta$ and thus are non-stationary throughout the training. The non-stationarity poses several challenges for the CEM. First, the stopping condition $\hat{q}_\alpha^{\phi_0}$ varies with $\pi_\theta$ and has to be re-estimated every iteration[1]. Second, the high-risk contexts $C$ (which correspond to the lowest returns) may vary as the agent evolves; and if the CEM learns to only sample a strict subset of the context space, then it may miss such changes in the high-risk contexts.

We address both issues using reference samples: every iteration, we sample *two* batches of contexts – $\{C_{\phi,i}\}_{i=1}^{N_s}$ from the current context distribution $D_\phi$ and $\{C_{o,i}\}_{i=1}^{N_o}$ from the original distribution $D_{\phi_0}$. The reference contexts provide an important regularization: they guarantee continual exposure to the whole context space, in case that the high-risk contexts vary. In addition, the reference samples were empirically found to stabilize the estimation of $\hat{q}_\alpha^{\phi_0}$ (Line 11 in Algorithm 1).

Consider the two batches of context-trajectory pairs, and denote the estimated return quantile $\hat{q}_\alpha = \hat{q}_\alpha(\{R(\tau_i)\}_{i=1}^{N_o})$. We can estimate the CVaR policy gradient, using the notation $\forall 1 \leq i \leq N_o + N_s$:

$$C_i = \begin{cases} C_{o,i} & \text{if } 1 \leq i \leq N_o \\ C_{\phi,i-N_o} & \text{if } N_o + 1 \leq i \leq N_o + N_s \end{cases}, \text{by}$$

$$\nabla_\theta \hat{J}_\alpha(\pi_\theta) = \frac{1}{\alpha(N_o + N_s)} \sum_{i=1}^{N_o+N_s} w_i \cdot \mathbf{1}_{R(\tau_i) \leq \hat{q}_\alpha} \left( R(\tau_i) - \hat{q}_\alpha \right) \sum_{t=0}^T \nabla_\theta \log \pi_\theta(a_{i,t}; s_{i,t}), \quad (6)$$

where $w_i = 1$ for $1 \leq i \leq N_o$ and $w_i = D_{\phi_0}(C_i)/D_{\phi^*}(C_i)$ for $N_o + 1 \leq i \leq N_o + N_s$.

Note that if the policy learning scale is slower than that of $\phi$, the target context distribution $P_{\phi_0,\alpha}^{\pi_\theta}$ is effectively stationary in the $\phi$-optimization problem. In that case, according to de Mello and Rubinstein [2003], the CEM will converge to the KL-divergence minimizer $\phi^*$ of (5).

---

[1]For the sake of coherent notations, we presented the CEM with the quantile objective $q_\alpha(R|\pi_\theta)$. In fact, the standard CEM is usually defined with a constant numeric objective $q_0 \in \mathbb{R}$ rather than a quantile; hence, as shown in Algorithm 2 in the appendix, the standard CEM does not require any quantile estimation at all.

**Sample efficiency in practice**: Proposition 1 guarantees an $\alpha^{-1}$-increase in sample efficiency when using an accurate quantile estimate $\hat{q}_\alpha = q_\alpha(R|\pi_\theta)$ and sampling exactly from $P^{\pi_\theta}_{\phi_0,\alpha}$. The latter condition is equivalent to the CE-sampler reaching its objective $D_{KL}(P^{\pi_\theta}_{\phi_0,\alpha} \,\|\, P^{\pi_\theta}_\phi) = 0$. In practice, $P^{\pi_\theta}_{\phi_0,\alpha}$ can only be approximated, and the sample efficiency is increased – but by a smaller factor than $\alpha^{-1}$. Appendix D.3 demonstrates the increased sample size exploited by CeSoR in our experiments.

If $\hat{q}_\alpha \neq q_\alpha(R|\pi_\theta)$, the quantile estimation error may theoretically lead to unbounded IS weights (see Appendix B). Practically, we address this by clipping the weights (as mentioned in Section 5), and by constraining the family of permitted distributions $\{D_\phi\}_\phi$ to have a constant support independently of $\phi$. A side-effect is a function approximation error of the family $\{D_\phi\}_\phi$, as $D_{\phi^*}(C)P^{\pi_\theta}_C(\tau)$ cannot replicate the tail distribution $P^{\pi_\theta}_{\phi_0,\alpha}(C,\tau)$ to achieve the full $\alpha^{-1}$-increase in sample efficiency.

Another limitation in the expressiveness of $P^{\pi_\theta}_\phi(C,\tau) = D_\phi(C)P^{\pi_\theta}_C(\tau)$ occurs when the context $C$ only controls part of the environment randomness in $P^{\pi_\theta}_C(\tau)$. As an extreme example in the Guarded Maze, after $\pi_\theta$ already learns to avoid the short path, the context (guard cost) does not affect the outcome at all anymore. Indeed, Figure 10a in the appendix shows that high guard costs are sampled in the beginning; then, once the short path is avoided, the sampler gradually falls back to the original context distribution. Note that in this example, the invariance to $C$ began after the learning was essentially done, hence the CEM did play its part effectively.

Finally, note that the soft risk creates an intentional bias in the gradient estimate (to overcome the blindness to success). As a result, in the first phase of training ($\alpha' \gg \alpha$), only a few trajectories are overlooked every iteration. As $\alpha'$ approaches $\alpha$, the number of overlooked trajectories increases, and so is the importance of over-sampling the tail. In the final steady-state phase ($\alpha' = \alpha$), the sample inefficiency is most severe, the soft risk produces no further biases, and the CEM helps CeSoR to reduce the high variance in the policy gradient estimation.

**The harmony between the soft risk and the CEM**: Soft risk has the inherent side effect of reducing the risk aversion. In the Guarded Maze, for example, as demonstrated in Section 5.1, soft risk alone leads to learning the short path (instead of the risk-averse long path). Fortunately, the CEM reduces this side effect. In that sense, the two mechanisms complement each other: $\alpha' > \alpha$ allows the *optimizer* to learn policies with high returns, while the CE *sampler* still preserves the risk aversion – as illustrated in Figure 3. This connection stands in addition to the independent motivations of the two mechanisms, as discussed above.

**Baseline optimizer**: CeSoR can be implemented on top of any CVaR-PG method as a baseline (Line 15). We use the standard GCVaR [Tamar et al., 2015b], which guarantees asymptotic convergence under certain regularity conditions. Appendix C shows that these guarantees hold for CeSoR as well, when implemented on top of GCVaR. Other CVaR-PG baselines can also be used, such as the TRPO-based algorithm of Rajeswaran et al. [2017]. However, such methods often include heuristics that introduce additional gradient estimation bias (to reduce variance), and thus do not necessarily guarantee the same theoretical convergence.

## 5    Experiments

We conduct experiments in 3 different domains. We implement **CeSoR** on top of a standard CVaR-PG method, which is also used as a risk-averse baseline for comparison. Specifically, we use the standard **GCVaR** baseline [Tamar et al., 2015b], which guarantees convenient convergence properties (see Appendix C) and is simple to implement and analyze. We also use the standard policy gradient (**PG**) as a risk-neutral baseline. We stress that the comparison to PG is only intended to present the mean-CVaR tradeoff, while each method legitimately optimizes its own objective. Appendix H also compares CeSoR to risk-neutral and risk-averse Distributional RL algorithms.

In all the experiments, all agents are trained using Adam [Diederik P. Kingma, 2014], with a learning rate selected manually per benchmark and $N = 400$ episodes per training step. Every 10 steps we run validation episodes, and we choose the final policy according to the best validation score (best mean for PG, best CVaR for GCVaR and CeSoR). For CeSoR, unless specified otherwise, $\nu = 20\%$ of the trajectories per batch are drawn from the original distribution $D_{\phi_0}$; $\beta = 20\%$ are used for the CE update; and the soft risk level reaches $\alpha$ after $\rho = 80\%$ of the training. As mentioned in Section 4, for numerical stability, we also clip the IS weights (Algorithm 1, Line 9) to the range $[1/5, 5]$.

Every policy is modeled as a neural network with $tanh$ activation on its middle layers and $softmax$ operator on its output, with temperature $1$ in training (i.e., network outputs are actions probabilities), and $0$ in validation and test (i.e., the max output is the selected action). We use a middle layer with 32 neurons in Section 5.2, 16 neurons in Section 5.3, and no middle layer (linear model) in Section 5.1.

In each of the 3 domains, the experiments required a running time of a few hours on an Ubuntu machine with eight i9-10900X CPU cores. In addition to these RL-related experiments, Appendix D presents dedicated experiments for the independent CE module.

## 5.1 The Guarded Maze

**Benchmark:** The Guarded Maze benchmark is defined in Section 1. For the experiments, we set a target risk level of $\alpha = 0.05$, and train each agent for $n = 250$ steps with the parameters described above. The CEM controls $C$ through $\phi = (\phi_1, \phi_2)$, where $\phi_0 = (0.2, 32)$ as mentioned above, and updates $\phi_1, \phi_2$ using the weighted means of $C_1$ and $C_2$, respectively. As an ablation test, we add two partial variants of our CeSoR: **CeR** (with CE, without $\alpha$-scheduling) and **SoR** (with scheduling, without CE). See more details in Appendix E.1.

**Results:** Figure 1a summarizes the test scores, and Figure 1d illustrates a sample episode. PG learned the short path, maximizing the average but at the cost of poor returns whenever charged by the guard. CeSoR, on the other hand, successfully learned to follow the CVaR-optimal long path. GCVaR, which also aimed to maximize the CVaR, failed to do so.

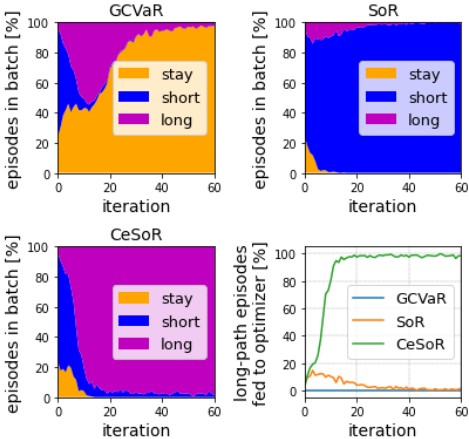

Figure 4: GCVaR, SoR, CeSoR: %-episodes that did not reach the target ("stay"), or reached it through the short or the long path in the Guarded Maze. Bottom Right: %-long-paths among the trajectories fed to the optimizer. See more details in Figure 13.

As analyzed in Figure 4, throughout GCVaR training, the agent takes the long path in up to 50% of the episodes per batch, but none of these episodes is ever included in the bottom $\alpha = 5\%$ that are fed to the optimizer. Thus, GCVaR is entirely *blind* to the successful episodes and fails to learn the corresponding strategy. In fact, in most training steps, *all* the worst episodes of GCVaR reach neither the guard nor the target, leading to a constant return of $-32$, a tail barrier, and a zero loss-gradient.

CeR suffers from blindness to success just as GCVaR. SoR is exposed to the successful long-path episodes thanks to soft risk scheduling; however, due to the reduced risk-aversion, it fails to prefer the long path over the short one. Only CeSoR both observes the *"good" strategy* (thanks to soft risk scheduling) and judges it under *"bad" environment variations* (thanks to the CEM). Appendix E.2 presents a detailed analysis of the learning dynamics, the blindness to success and the learned policies. It is important to notice that standard optimization tweaks cannot bring GCVaR to learn the long path: a "warm-start" from a standard PG only encourages the short-path policy (as in SoR); and increased batch size $N$ does not expose the optimizer to the long path (see Theorem 1).

## 5.2 The Driving Game

**Benchmark:** The Driving Game is based on an inverse-RL benchmark used by Majumdar et al. [2017] and Singh et al. [2018]. The agent's car has to follow the leader (an "erratic driver") for 30 seconds as closely as possible without colliding. Every 1.5 seconds (i.e., 20 times per episode), the leader chooses a random action (independently of the agent): drive straight, accelerate, decelerate, change lane, or brake hard ("emergency brake"), with respective probabilities $\phi_0 = (0.35, 0.3, 0.248, 0.1, 0.002)$. We denote the sequence of leader actions by $C \in \{1, ..., 5\}^{20}$.

Every 0.5 seconds (60 times per episode), the agent observes its relative position and velocity to the leader, with a delay of 0.7 seconds (representing reaction time), as well as its own acceleration and steering direction. The agent chooses one of the five actions: drive in the same steering direction, accelerate, decelerate, turn left, or turn right. Changing lane is not an atomic action and has to be learned using turns. The rewards express the requirements to stay behind the leader, on the road, on the same lane, not too far behind and without colliding. See the complete details in Appendix F.1.

We set $\alpha = 0.01$, and train each agent for $n = 500$ steps. To initiate learning, for each agent we begin with shorter training episodes of 6 seconds and gradually increase their length. The CEM controls the leader's behavior through the probabilities $\phi = \{\phi_i\}_{i=1}^5$ described above.

**Results:** Figure 1b summarizes the test scores of the agents, where CeSoR presents a reduction of 28% in the CVaR cost in comparison to the baselines. GCVaR completely fails to learn a reasonable policy – losing in terms of CVaR even to the risk-neutral PG. Figure 5 shows that CeSoR learned an arguably-intuitive policy for risk averse driving: it keeps a safer distance from the leader, and uses the gas and the brake less frequently. This results in complete avoidance of the rare accidents occurring to PG, as demonstrated in Figure 1e. In Appendix D, we also see that by over-sampling turns and emergency brakes of the leader, the CEM manages to align the mean return of the training samples with the 1%-CVaR of the environment, and significantly increases the data efficiency.

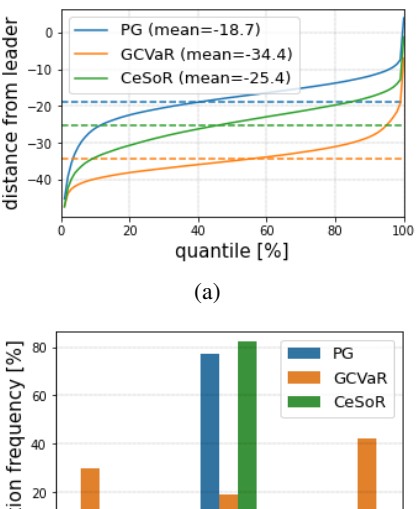

### 5.3 The Computational Resource Allocation Problem

**Benchmark:** Computational resource allocation in serving systems, and in particular the tradeoff between resource cost and serving latency, is an important challenge to both academia [Jiang et al., 2013, Tessler et al., 2022] and industry [Barr, 2018, Lunden, 2022]. In popular applications such as E-commerce and news, latency is most critical at times of peak loads [Garces, 2019], making CVaR a natural metric for risk-averse optimization. In our benchmark, the agent allocates servers to handle user requests, managing the tradeoff between servers cost and time-to-service (TTS). Requests arrive randomly with a constant rate, up to rare events that cause sudden peak

Figure 5: Over all the time-steps in all the test episodes in the Driving Game, the distribution of (a) the distance between the agent and the leader, (b) the agent actions. Evidently, CeSoR learns to keep more distance than the risk-neutral PG, and has a slightly less frequent use of the gas and the brake.

loads, whose frequency is controlled by the CE sampler. See Appendix G for more details.

**Results:** As shown in Figure 1c, CeSoR significantly improves the CVaR return, and does not compromise the mean as much as GCVaR. As demonstrated in Figure 1f, CeSoR learned to allocate a default of 5 servers and react to peak loads as needed, whereas GCVaR simply allocates 8 servers at all times. PG only allocates 4 servers by default, and thus its TTS is more sensitive to peak loads. Appendix G describes the complete implementation and detailed results, discusses the poor parameterization of $D_\phi$ in this problem and shows the robustness of CeSoR to that parameterization.

## 6 Summary and Future Work

We introduced CeSoR, a novel method for risk-averse RL, focused on efficient sampling and soft risk. In a variety of experimental domains, in comparison to a risk-averse baseline, CeSoR demonstrated higher CVaR metric, better sample-efficiency, and elimination of blindness to success – where the latter two were also analyzed theoretically.

There are certain limitations to CeSoR. First, we assume to have at least partial control over the training conditions, through a parametric family of distributions that needs to be selected. Second, CeSoR can be applied robustly on top of any CVaR-PG method, but is currently not applicable to non-PG methods. Since the limitations of CVaR-PG apply in other risk-averse methods as well (as we demonstrated for Distributional RL), future work may adjust CeSoR to such methods, as well as to other risk measures. Third, in terms of blindness to success and estimation variance, CeSoR shows both theoretical and empirical improvement – but is not proven optimal. Future work may look for optimal design of CEM or risk scheduling. Considering the current results and the potential extensions, we believe CeSoR may open the door for more practical applications of risk-averse RL.

**Acknowledgements:** This research was supported by the Israel Science Foundation (grant 2199/20).

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
