# Contents

# A Blindness to Success: Proof of Theorem 1

Theorem 1 considers the probabilistic event of a global blindness to success over $n$ consecutive training steps. We begin with a local blindness in a single training step.

**Lemma 1** (Local blindness to success). Let a risk level $\alpha \in (0, 1)$ and a CVaR-PG training step $m \geq 1$, and let $\beta \in (\alpha, 1)$. Denote $A = \left\{ \{\tau_{m,i}\}_{i=1}^N \in \mathcal{T}^N \mid q_\beta^{\pi_m} < \hat{q}_\alpha(\{R(\tau_{m,i})\}_{i=1}^N) \right\}$. Then,

$$\mathbb{P}(A) \leq e^{-\frac{N(\beta-\alpha)^2}{2\beta(1-\beta)}} \leq e^{-2N(\beta-\alpha)^2}$$

*Proof.* Denote $R_i = R(\tau_{m,i})$, $\chi_i^q = \mathbf{1}_{R_i > q}$, and $\chi_i = \chi_i^{q_\beta^{\pi_m}}$. Note that $\chi_i \sim Bernoulli(1 - \beta)$. Also denote the percent of high-return trajectories by $n_q = \sum_{i=1}^N \chi_i^q / N$ and $n^* = n_{q_\beta^\pi}$. Since $\hat{q}_\alpha(\{R_i\}_{i=1}^N) = \min\left\{ q \mid \frac{|\{i \mid R_i \leq q\}|}{N} \geq \alpha \right\} = \min\left\{ q \mid \frac{|\{i \mid R_i > q\}|}{N} < 1 - \alpha \right\} = \min\left\{ q \mid \frac{1}{N} \sum_{i=1}^N \chi_i^q < 1 - \alpha \right\}$, we have $q_\beta^\pi < \hat{q}_\alpha(\{R_i\}) \Leftrightarrow n^* \geq 1 - \alpha$, i.e., $A = (n^* \geq 1 - \alpha)$.

Since $\mathbb{P}(0 \leq \chi_i \leq 1) = 1$, $E[\chi_i] = 1 - \beta$ and the Bernoulli $\chi_i$ are sub-Gaussian with variance factor $\sigma^2 = 1/4$, by Hoeffding inequality [Hoeffding, 1994] we obtain

$$\mathbb{P}(A) = \mathbb{P}(n^* \geq 1 - \alpha) = \mathbb{P}(n^* - E[n^*] \geq \beta - \alpha) \leq e^{-\frac{N^2(\beta-\alpha)^2}{2\sum_i 1/4}} = e^{-2N(\beta-\alpha)^2}.$$

$\square$

Note that Lemma 1 does not depend on a tail-barrier: it simply implies that since a CVaR-PG algorithm focuses on the worst $\alpha$ trajectories in every batch, we do not expect trajectories with high returns $R(\tau_{m,i}) > q_\beta^{\pi_m}$ to be fed to the optimizer. Still, in general, even if high-return trajectories are ignored, the CVaR-PG can learn to avoid low-return trajectories with $R(\tau_{m,i}) < q_\beta^{\pi_m}$. The tail barrier prevents this learning, since there are no returns strictly lower than $q_\beta^{\pi_m}$ – all the tail identically equals $q_\beta^{\pi_m}$. Since there are no worse trajectories to learn from, and better trajectories are ignored, this brings the training to a deadlock, as stated by Theorem 1.

*Proof of Theorem 1 (stated in Section 3.1).* All probabilities below are conditioned on the event of $\pi_{m_0}$ having a $\beta$-tail barrier. Thus, we simplify the notation to $\mathbb{P}(\cdot) = \mathbb{P}(\cdot \mid \pi_{m_0}$ has $\beta$-tail barrier$)$.

Denote by $\mathcal{S} = \left\{ \left( \{\tau_{m,i}\}_{i=1}^N, \pi_m \right) \right\}_{m=m_0}^{m_0+n-1} \in (\mathcal{T}^N \times \Pi)^n$ the sequence of trajectory batches and policies, and by $R_m = \{R(\tau_{m,i})\}_{i=1}^N$ the returns on step $m$. Also denote for simplicity $\mathcal{B} = \mathcal{B}_{\alpha,\beta}^{m_0,n}$. We are interested in the probability of the event that there is no global blindness (Definition 2):

$$\neg\mathcal{B} = \neg\mathcal{B}_{\alpha,\beta}^{m_0,n} = \left\{ \mathcal{S} \mid \exists m_0 \leq m < m_0 + n : q_\beta^{\pi_{m_0}} < \hat{q}_\alpha(R_m) \right\}.$$

Define the event of blindness at step $m$, along with an unchanged policy: $A_m = \left\{ \mathcal{S} \mid \pi_m = \pi_{m_0} \wedge \hat{q}_\alpha(R_m) \leq q_\beta^{\pi_{m_0}} \right\}$. Note that $\bigcap_{m=m_0}^{m_0+n-1} A_m \subseteq \mathcal{B}$, hence

$$\mathbb{P}(\neg\mathcal{B}) \leq 1 - \mathbb{P}\left( \bigcap_{m=m_0}^{m_0+n-1} A_m \right) = 1 - \prod_{m=m_0}^{m_0+n-1} \mathbb{P}(A_m | A_{m_0}, ..., A_{m-1}).$$

Thus, to complete the proof, we show below that $\mathbb{P}(A_m | A_{m_0}, ..., A_{m-1}) \geq 1 - \delta$, where $\delta = e^{-2N(\beta-\alpha)^2}$, hence $\mathbb{P}(\neg\mathcal{B}) \leq 1 - (1 - \delta)^n \leq 1 - (1 - n\delta) = n\delta$.

For $m = m_0$, we have immediately $\pi_m = \pi_{m_0}$, and from Lemma 1 $\mathbb{P}(q_\beta^{\pi_{m_0}} < \hat{q}_\alpha(R_{m_0})) \leq \delta$. For $m_0 + 1 \leq m \leq m_0 + n - 1$, assume that $A_{m_0}, ..., A_{m-1}$ hold. In particular, $\hat{q}_\alpha(R_{m-1}) \leq q_\beta^{\pi_{m_0}}$, $\pi_{m-1} = \pi_{m_0}$ and $\pi_{m-1}$ has a $\beta$-tail barrier. Now consider the $m - 1$ training batch: for every trajectory $1 \leq i \leq N$, if $R_{m-1,i} > \hat{q}_\alpha(R_{m-1})$, then $\mathbf{1}_{R_{m-1,i} \leq \hat{q}_\alpha(R_{m-1})} = 0$; otherwise, $R_{m-1,i} \leq \hat{q}_\alpha(R_{m-1}) \leq q_\beta^{\pi_{m_0}}$, that is, $R_{m-1,i} = q_{\beta'}^{\pi_{m_0}}$ for some $\beta' \leq \beta$, and by the barrier property $R_{m-1,i} = q_\beta^{\pi_{m_0}}$ and thus $R_{m-1,i} - \hat{q}_\alpha(R_{m-1}) = 0$. Hence, the gradient in Equation (3) is 0, the

policy update vanishes, and we obtain $\pi_m = \pi_{m-1} = \pi_{m_0}$. Then again, according to Lemma 1 (and since $R_m$ and $R_{m_0}$ are drawn from the same distribution corresponding to $\pi_m = \pi_{m_0}$), we have $\mathbb{P}(q_\beta^{\pi_{m_0}} < \hat{q}_\alpha(R_m)) = \mathbb{P}(q_\beta^{\pi_{m_0}} < \hat{q}_\alpha(R_{m_0})) \leq \delta$, as required. $\qquad\square$

Note that the factor $n$ may become quite negligible when the barrier is wider than $\alpha$: if $n = 10^6, \alpha = 0.05, \beta = 0.25, N = 400$, for example, we still have $\mathbb{P}(\neg\mathcal{B}_{\alpha,\beta}^{m_0,n}) < 10^{-7}$. Indeed, the blindness occurs with significantly smaller barriers than the $\beta = 0.9$ demonstrated in the Guarded Maze in Appendix E.2. Note that the momentum term of the Adam algorithm [Diederik P. Kingma, 2014], while preventing the policy update from completely vanishing, was empirically insufficient to overcome the barrier in the Guarded Maze. This should not come as a surprise, since the momentum comes from previous gradients that *encouraged* the strategies of the barrier and brought them into the tail in the first place.

## B  Variance Reduction: Proof of Proposition 1

*Proof.* Define $H(C, \tau) = \alpha^{-1}\mathbf{1}_{R(\tau)\leq\hat{q}_\alpha} \left(R(\tau) - \hat{q}_\alpha\right) \nabla_\theta \sum_t \log \pi_\theta(a_t; s_t)$, such that the CVaR PG can be written as $\nabla_\theta \hat{J}_\alpha \left(\{C_i, \tau_i\}_{i=1}^N; \pi_\theta\right) = \frac{1}{N} \sum_{i=1}^N w(C_i, \tau_i) H(C_i, \tau_i)$, where $w(C, \tau) = \frac{P_{\phi_0}^{\pi_\theta}(C,\tau)}{P_{\phi_0,\alpha}^{\pi_\theta}(C,\tau)}$ is the IS weighting that accounts for the modified sample distribution. Since $C, \tau \sim P_{\phi_0,\alpha}^{\pi_\theta}$, we have $R(\tau) \leq q_\alpha(R|\pi_\theta)$ almost surely; and along with the assumption $\hat{q}_\alpha = q_\alpha(R|\pi_\theta)$, we obtain

$$w(C, \tau) = \frac{P_{\phi_0}^{\pi_\theta}(C,\tau)}{P_{\phi_0,\alpha}^{\pi_\theta}(C,\tau)} = \frac{P_{\phi_0}^{\pi_\theta}(C,\tau)}{\alpha^{-1}\mathbf{1}_{R(\tau)\leq q_\alpha(R|\pi_\theta)}P_{\phi_0}^{\pi_\theta}(C,\tau)} = \alpha.$$

The assumption $\hat{q}_\alpha = q_\alpha(R|\pi_\theta)$, when applied to Equation (3), also guarantees that $\nabla_\theta \hat{J}_\alpha$ is an unbiased gradient estimator for both sample distributions $P = P_{\phi_0}^{\pi_\theta}$ and $P = P_{\phi_0,\alpha}^{\pi_\theta}$: $\mathbb{E}_{C_i,\tau_i\sim P}[\nabla_\theta \hat{J}_\alpha \left(\{C_i, \tau_i\}_{i=1}^N; \pi_\theta\right)] = \nabla_\theta J_\alpha (\pi_\theta)$. Its variance over $N$ i.i.d samples is $\mathrm{Var}_{C_i,\tau_i\sim P}[\nabla_\theta \hat{J}_\alpha \left(\{C_i, \tau_i\}_{i=1}^N; \pi_\theta\right)] = \frac{1}{N}\mathrm{Var}_{C,\tau\sim P}[\nabla_\theta \hat{J}_\alpha (C, \tau; \pi_\theta)]$. Denoting $g := \nabla_\theta J_\alpha (\pi_\theta)$, we obtain:

$$\mathrm{Var}_{C,\tau\sim P_{\phi_0,\alpha}^{\pi_\theta}}[\nabla_\theta \hat{J}_\alpha (C, \tau; \pi_\theta)]$$
$$=\mathbb{E}_{C,\tau\sim P_{\phi_0,\alpha}^{\pi_\theta}}[w(C,\tau)^2 H(C,\tau)^2] - g^2$$
$$=\mathbb{E}_{C,\tau\sim P_{\phi_0}^{\pi_\theta}}[w(C,\tau) H(C,\tau)^2] - g^2$$
$$=\alpha \cdot \mathbb{E}_{C,\tau\sim P_{\phi_0}^{\pi_\theta}}[H(C,\tau)^2] - g^2$$
$$\leq\alpha \cdot \left(\mathbb{E}_{C,\tau\sim P_{\phi_0}^{\pi_\theta}}[H(C,\tau)^2] - g^2\right)$$
$$=\alpha \cdot \mathrm{Var}_{C,\tau\sim P_{\phi_0}^{\pi_\theta}}[\nabla_\theta \hat{J}_\alpha (C, \tau; \pi_\theta)],$$

which completes the proof. $\qquad\square$

Note that if $\hat{q}_\alpha \neq q_\alpha(R|\pi_\theta)$, the term $\mathbf{1}_{R(\tau)\leq\hat{q}_\alpha}$ in the denominator may vanish and the IS weight $w(\tau, C)$ may become unbounded. To overcome this issue when using our CE-sampler (described in Section 4), we constrain the family of distributions $\{P_\phi^{\pi_\theta}\}_\phi$ such that the sample distribution $P_\phi^{\pi_\theta}$ always has the same support as the original distribution $P_{\phi_0}^{\pi_\theta}$ (even though this eliminates the possibility of an exact tail sampling $P_\phi^{\pi_\theta} = P_{\phi_0,\alpha}^{\pi_\theta}$). In addition, in the experiments of Section 5 we clip the IS weights directly.

## C  Gradient Estimation Bias and CeSoR Convergence

The gradient estimator of Equation (3) is biased due to the biasedness of the empirical quantile. However, Tamar et al. [2015b] show that the gradient estimator is still consistent, and bound its bias by $\mathcal{O}(N^{-1/2})$. Lemma 2 below proves that a similar result holds for CeSoR – despite the CEM and

the risk scheduling. Given Lemma 2, CeSoR's convergence is a direct application of Theorem 5 in Tamar et al. [2015b], as stated below. The soft-risk scheduling $\alpha'$ introduces additional transient bias to the CVaR gradient estimate when $\alpha' > \alpha$, but this bias vanishes in the last steady-state $1 - \rho$ steps when $\alpha' = \alpha$; hence, we can safely assume consistency of CeSoR's gradient estimate, and focus our asymptotic convergence analysis on the steady-state phase.

Formally, in terms of Section 2, assume that the update step includes a $\ell_p$ projection $\Gamma$ to a compact set with a smooth boundary: $\theta_{m+1} = \Gamma(\theta_m + \eta_m \nabla_\theta \hat{J}_\alpha)$; and that the learning rate $\eta_m$ satisfies $\sum_{m=0}^\infty \eta_m = \infty$, $\sum_{m=0}^\infty \eta_m^2 < \infty$ and $\sum_{m=0}^\infty \eta_m \left| E\left[\nabla_\theta \hat{J}_\alpha\right] - \nabla_\theta J_\alpha \right| < \infty$ w.p. 1. In addition, denote by $\mathcal{K}$ the set of all asymptotically-stable equilibria of the ODE $\dot{\theta} = \Gamma(\nabla_\theta J_\alpha(R; \pi_\theta))$.

**Theorem 2** (Convergence of CeSoR). Assume that for any $\phi$, the sample distribution $D_\phi$ of Algorithm 1 has the same support as the original distribution $D_{\phi_0}$. Then, under the smoothness assumptions specified in Appendix C.1, and the projection and learning rate assumptions specified above, the sequence of policy parameters $\{\theta_m\}$ generated by Algorithm 1 converges almost surely to $\mathcal{K}$.

Theorem 2 relies on similar assumptions to Tamar et al. [2015b], two of them are of particular interest in our context. First, the rewards are assumed to be continuous. Second, in the gradient estimator, the baseline is assumed to be a consistent estimator of the returns $\alpha$-quantile. Hence, while CeSoR is compatible with any CVaR-PG method, the current derivation of theoretical convergence guarantees only holds for PG methods with a consistent gradient estimate.

### C.1 Gradient Estimation Bias

The gradient estimator of the standard CVaR PG may be inconsistent and unboundedly-biased, unless the return baseline is a consistent estimator of the $\alpha$-quantile of the returns [Tamar et al., 2015b]. Thus, we rely on the empirical quantile baseline $\hat{q}_\alpha$ used in Equation (3), which is a consistent (though biased) estimator of the true quantile. Given certain smoothness assumptions, Tamar et al. [2015b] bound the resulted bias of the gradient estimator $E\left[\nabla_\theta \hat{J}_\alpha\right] - \nabla_\theta J_\alpha$ (as defined in Equations (2),(3)). Lemma 2 guarantees that under the same assumptions, despite the modified sampling by the CEM, the same bias bounds apply to CeSoR.

We first specify the smoothness assumptions. Note that Tamar et al. [2015b] consider $\nabla_\theta \log f_{s|a}(s|a, \theta)$ in their calculations (or in their notation: $\nabla_\theta \log f_{X|Y}(X|Y, \theta)$). In RL applications, given the action $a$, the next-state distribution is independent of the policy $\pi_\theta$, and this gradient vanishes. We accordingly ignore this term in the calculations, which simplifies the assumptions and the analysis. The remaining assumptions mostly consider the smoothness of the rewards, and in particular do not hold in the case of discrete rewards as discussed in Section A.

**Assumption 1** (Smoothness assumptions). For any policy $\pi_\theta$, the return $R$ is a continuous random variable; and $\nabla_\theta q_\alpha(R; \pi_\theta)$, $\nabla_\theta J_\alpha(\pi_\theta)$ and $\nabla_\theta \log \pi_\theta(a)$ (for any $a$) are well defined and bounded.

**Lemma 2** (Gradient estimation bias bound). In Algorithm 1 with a batch size $N$, consider a certain step $m \geq \rho M$, and assume that the underlying PG follows Equation (3) (or Equation (6)). In addition, assume that for any $\phi$, the sample distribution $D_\phi$ of Algorithm 1 has the same support as the original distribution $D_{\phi_0}$. Then, under Assumption 1, $E\left[\nabla_\theta \hat{J}_\alpha\right] - \nabla_\theta J_\alpha = \mathcal{O}(N^{-1/2})$.

*Proof.* We follow the steps of the proof of Theorem 4 in Tamar et al. [2015b] with the following modifications. First, we take the gradient expectations with respect to the CE sampling distribution $D_\phi$ rather than the original distribution $D_{\phi_0}$. Second, the empirical quantile $\hat{q}_\alpha$ is calculated in Algorithm 1 using a reduced sample size $N_o = \lfloor \nu N \rfloor < N$. Note that the estimator $\hat{q}_\alpha$ relies on samples drawn from $D_{\phi_0}$, hence is not otherwise affected by the CEM.

Denote by $D_{\phi_i}$ the distribution from which was drawn $C_i$, i.e., $\phi_i = \phi_0$ for $i \leq N_o$ and $\phi_i = \phi$ for $i > N_o$. Since $m \geq \nu N$, according to Line 13 in Algorithm 1 we have $\alpha' = \alpha$. Denoting by $q_\alpha$ the true $\alpha$-quantile of the returns, we have

$$\nabla_\theta J_\alpha(R; \pi_\theta) = E_{\{\phi_i\}_{i=1}^N} \left[ \frac{1}{\alpha N} \sum_{i=1}^N w_i \mathbf{1}_{R_i \leq q_\alpha} (R_i - q_\alpha) \nabla_\theta \log \pi_\theta(\tau_i) \right] \tag{7}$$

We now substitute $w_i = \frac{D_{\phi_0}(C_i)}{D_{\phi_i}(C_i)}$, which is finite due to the assumption that $D_{\phi_i}$ has the same support as $D_{\phi_0}$. Using the notation $E_{\phi_0}[\cdot] = E_{C \sim D_{\phi_0}, \tau \sim P_C^{\pi_\theta}}[\cdot]$ and $\pi_\theta(\tau_i) = \Pi_t \pi_\theta(a_{i,t}; s_{i,t})$, we obtain

$$\left| E_{C_i \sim D_{\phi_i}, \tau_i \sim P_{C_i}^{\pi_\theta}} \left[ \nabla_\theta \hat{J}_\alpha(\{\tau_i\}; \pi_\theta) \right] - \nabla_\theta J_\alpha(\pi_\theta) \right|$$

$$\leq E_{C_i \sim D_{\phi_i}, \tau_i \sim P_{C_i}^{\pi_\theta}} \left[ \frac{1}{\alpha N} \sum_{i=1}^{N} \frac{D_{\phi_0}(C_i)}{D_{\phi_i}(C_i)} \left| \nabla_\theta \log \pi_\theta(\tau_i) \left( \mathbf{1}_{R_i \leq \hat{q}_\alpha} (R_i - \hat{q}_\alpha) - \mathbf{1}_{R_i \leq q_\alpha} (R_i - q_\alpha) \right) \right| \right]$$

$$= E_{\phi_0} \left[ \frac{1}{\alpha N} \sum_{i=1}^{N} \left| \nabla_\theta \log \pi_\theta(\tau_i) \left( \mathbf{1}_{R_i \leq \hat{q}_\alpha} (R_i - \hat{q}_\alpha) - \mathbf{1}_{R_i \leq q_\alpha} (R_i - q_\alpha) \right) \right| \right]$$

$$= E_{\phi_0} \left[ \frac{1}{\alpha N} \sum_{i=1}^{N} \left| \nabla_\theta \log \pi_\theta(\tau_i) \left( (\mathbf{1}_{R_i \leq \hat{q}_\alpha} - \mathbf{1}_{R_i \leq q_\alpha}) (R_i - \hat{q}_\alpha) + \mathbf{1}_{R_i \leq q_\alpha} ((R_i - \hat{q}_\alpha) - (R_i - q_\alpha)) \right) \right| \right]$$

$$\leq E_{\phi_0} \left[ \frac{1}{\alpha N} \sum_{i=1}^{N} \left| \nabla_\theta \log \pi_\theta(\tau_i) (\mathbf{1}_{R_i \leq \hat{q}_\alpha} - \mathbf{1}_{R_i \leq q_\alpha}) (R_i - \hat{q}_\alpha) \right| \right]$$

$$+ E_{\phi_0} \left[ \frac{1}{\alpha N} \sum_{i=1}^{N} \left| \nabla_\theta \log \pi_\theta(\tau_i) \mathbf{1}_{R_i \leq q_\alpha} (q_\alpha - \hat{q}_\alpha) \right| \right]$$

$$(8)$$

From this point, the proof is mostly identical to Theorem 4 in Tamar et al. [2015b]. Namely, the first term is $o(N^{-1/2})$ according to Hong and Liu [2009], given Assumption 1; and since $\hat{q}_\alpha$ is estimated using $\nu N$ samples, we have $|q_\alpha - \hat{q}_\alpha| = \mathcal{O}((\nu N)^{-1/2}) = \mathcal{O}(N^{-1/2})$ in probability (note that $\nu$ is constant, e.g., $\nu = 0.2$ or $\nu = 0.5$ in the experiments of Section 5). Together, the whole expression is $\mathcal{O}(N^{-1/2})$ as required.

$\square$

# D   The Cross Entropy Module: Extended Discussion

The Cross Entropy Method (CEM) with non-stationary score function has a major role in CeSoR. The CEM code is implemented and available as an independent module [Greenberg, 2022]. Below we present an analysis of the CEM empirical results over both a dedicated toy problem (which tests the CEM independently of CeSoR) and as part of CeSoR in the benchmarks of Section 5.

## D.1   The CEM Algorithm

For clarity, we first provide the pseudo-code for the general CEM algorithm. This version repeatedly generates samples from the tail of the distribution $D_{\phi_0}$. A similar version [de Boer et al., 2005] would stop once $q_\beta \left( \{R(x_i)\}_{i=1}^{N} \right) \leq q$ (as it means that at least $\beta N$ samples are already beyond $q$), and use all the recent samples $R(x_i) \leq q$ to estimate the probability of the "rare event" $R(X) \leq q$.

Note that unlike CeSoR, Algorithm 2 relies on a constant mapping $R(x)$ and a constant target $q$. Our CEM version in CeSoR, as implemented in our code and presented in Algorithm 1, supports a quantile-target $\alpha$ with respect to a return mapping $R$ that varies dynamically with the learning agent.

**Algorithm 2:** The Cross Entropy Method for Sampling

1 **Input**: distribution $D_{\phi_0}$; score function $R$; target level $q$; batch size $N$; update selection rate $\beta$.

2 $\phi \leftarrow \phi_0$
3 **while** *true* **do**
    // Sample
4     Sample $x \sim D_\phi^N$
5     $w_i \leftarrow D_{\phi_0}(x_i)/D_\phi(x_i) \quad (1 \leq i \leq N)$
6     Print $x$
    // Update
7     $q' \leftarrow \max\left(q,\ q_\beta\left(\{R(x_i)\}_{i=1}^N\right)\right)$
8     $\phi \leftarrow \operatorname{argmax}_{\phi'} \sum_{i=1}^N w_i \mathbf{1}_{R(x_i) \leq q'} \log D_{\phi'}(x_i)$

### D.2 Sample Distribution

The goal of the CEM is to align the sample distribution with the bottom-$\alpha$ percent of the reference distribution. Note that given a parametric family of distributions $D_\phi$ with a limited expressiveness, a perfect alignment is not always possible. For example, if the CEM controls the mean of an exponential distribution $C \sim Exp(\phi)$, and the returns decrease with $c$, then the lower quantiles of the returns correspond to $C \geq q_\alpha(C)$. However, no value of $\phi$ could eliminate the lower values $C \in [0, q_\alpha]$ – but could merely assign more probability density to higher values. Even when the family of distributions is expressive enough, the CEM has to learn the desired sample distribution without any prior knowledge about the meaning of the parameters that it controls. In particular, it cannot know in advance in which direction each parameter may affect the agent return, what the size of the effect would be, and how it would change during the training.

Formally, the objective of the CEM is often defined as minimization of the KL-divergence between the sample distribution and the desired tail of the reference distribution [Dambreville, 2006]. Indeed, this objective is well-defined even if the expressiveness of $D_\phi$ does not allow a perfect alignment.

In this section, we focus on the comparison between the mean and the CVaR of the sample distribution and the reference distribution of the returns. Specifically, while both distributions begin with the same mean and CVaR, we hope that the sample mean would align with the reference CVaR as quickly as possible.

First, we consider a toy problem with a static reference distribution and no RL environment. The parametric family of distributions is $C \sim Beta(2\phi, 2 - 2\phi)$ (such that $E[C] = \phi$), and the reference distribution corresponds to $\phi_0 = 0.5$, which results in the uniform distribution $Beta(1, 1) = U(0, 1)$. We are interested in the bottom $\alpha = 10\%$ of the reference distribution, i.e., $U(0, 0.1)$. We run the CEM for $n = 10$ steps with $N = 1000$ samples per step, $\nu = 20\%$ of them are drawn from the original reference distribution, and update $\phi$ using the mean of the lower $\beta = 50\%$ samples. Note that generally in this work, $C$ is the context or configuration of an environment that produces returns; in this toy example, we do not have an RL environment and we simply define $R(C) = C$.

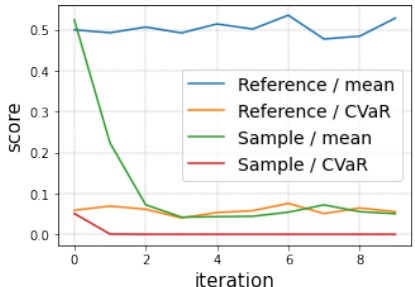

Figure 6: The converges of the CE sample mean to the reference $CVaR_{10\%}$ in the toy $Beta$ distribution problem.

The $CVaR_{10\%}$ of $C$ (or equivalently, the mean of $U(0, 0.1)$) is 0.05. Note that no value of $\phi$ can yield the distribution $U(0, 0.1)$, as the support of the $Beta$ distribution is always $(0, 1)$. Yet, as shown in Figure 6, the sample mean converges to the reference CVaR within mere 2 iterations, and remains around this level.

Figure 7 presents the same metrics for the experiments described in Section 5. In these cases, the reference returns distribution corresponds to the agent returns under the original environment. Note that this reference returns distribution is dynamic during the training, as it changes with the agent (and in certain benchmarks also with the episode length that increases throughout the training). Yet, in the Driving Game benchmark, for example, we see that the sample mean reasonably aligns with the reference CVaR throughout most of the training, even as both of them vary.

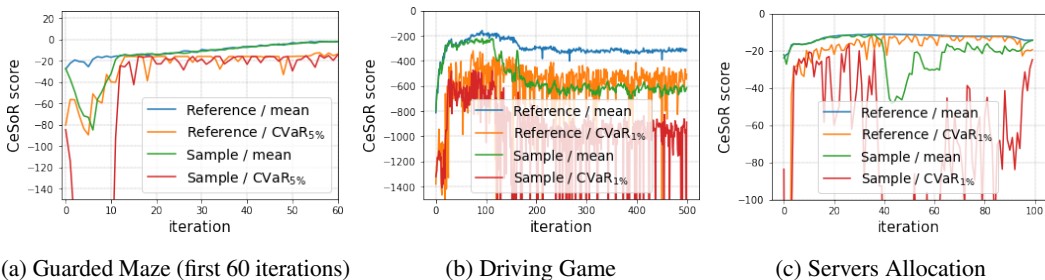

(a) Guarded Maze (first 60 iterations)     (b) Driving Game     (c) Servers Allocation

Figure 7: The mean and CVaR metrics of the CE sample distribution and the reference original distribution throughout the training of CeSoR over different benchmarks.

In the Guarded Maze, the sample mean also quickly converges into the reference CVaR. However, once the agent learns to avoid the short path, the CE sampler can no longer control the agent performance at all, and due to the regularizing reference samples, the sample distribution gradually goes back to the original one. This is a valid behavior, as the agent already learned to avoid the risk, and if for some reason it came back to the risky short path, the CE would simply learn again to focus on the risky configurations of the environment.

The Servers Allocation Problem takes the challenge of the CEM to the limit, as the target is $\alpha = 1\%$, the difficulty to the agent arrives in a non-smooth manner as rare and discrete events, and the given family of distributions (Binomial) has limitations in expressing the desired distribution. Specifically, we would like most of the sample episodes to include a peak event, but not more than one; whereas the Binomial distribution is not best-suitable for this. However, even as the CEM struggles to fit the reference $CVaR_{1\%}$ (Figure 7c), CeSoR is still shown to provide beneficial results (Section 5.3, Appendix G). This demonstrates the robustness of CeSoR to limitations and misspecification of the modeled family of distributions.

**Sensitivity to $\beta$:** As discussed in Sections 2 and 4, the smoothness parameter $\beta$ determines the minimal percent of data samples used for the update step in the CEM. We argue that CeSoR has a low sensitivity to the parameter $\beta$.

Intuitively, every iteration of the CEM focuses on the $\beta$-tail of the previous iteration (until reaching the $\alpha$-tail of the reference distribution). Theoretical analysis of the convergence rate is challenging, due to the limited expressiveness of $D_\phi$ and the non-stationary agent returns; yet, according to the qualitative intuition above, we expect exponential convergence to the tail, which applies even for high values of $\beta$. On the other hand, while low values of $\beta$ may increase the noise in the update step of the CEM, any noisy update could be corrected throughout the training. Note that Algorithm 1 uses the original context-distribution for a certain part of the samples of each batch; this guarantees that any update step is reversible, as CeSoR continues to be exposed to the complete context-space.

Empirically, we repeated the experiments of Section 5 with various values of $\beta \in [0.05, 0.3]$. In the Guarded Maze and the Driving Game, all the values of $\beta$ resulted in similar test returns; in addition, Figure 8 shows that the CEM successfully aligned the sample mean return with the reference CVaR, independently of $\beta$. The Servers Allocation Problem is more challenging for the CEM (as discussed above), making the sampler more sensitive to the parameter $\beta$, and in particular leading to a failure for $\beta = 0.3$. However, note that even under such a combination of poor algorithmic choices (Binomial

parameterization of $D_\phi$ and very high $\beta$), the failure of the CEM is easy to notice through Figure 8c (as the sample-mean fails to deviate from the reference-mean), and is easy to fix.

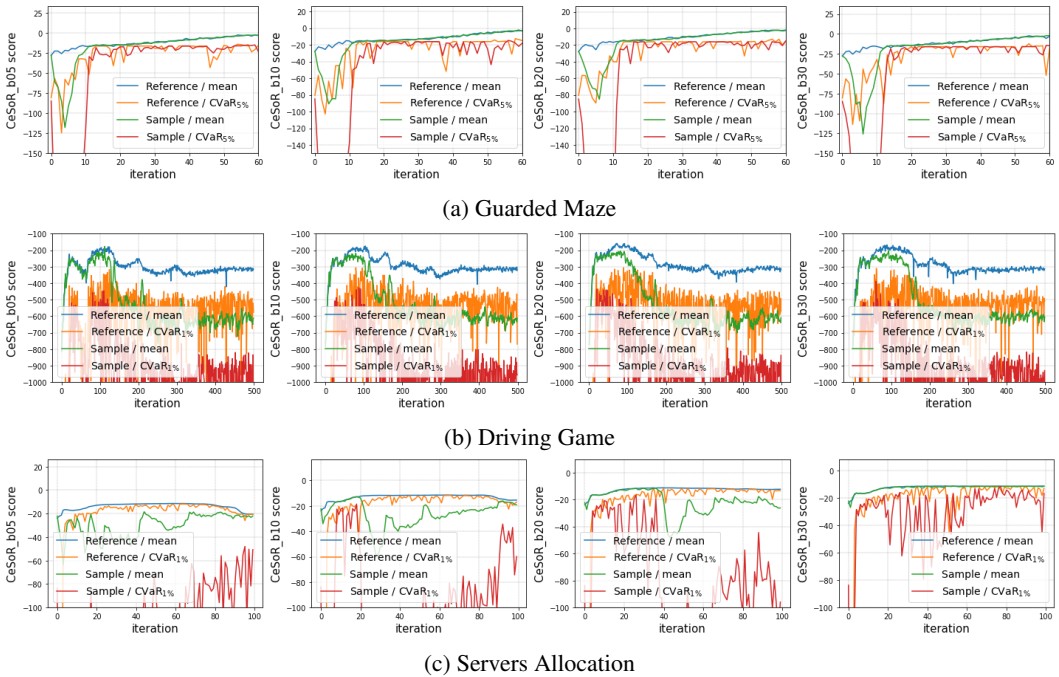

(a) Guarded Maze

(b) Driving Game

(c) Servers Allocation

Figure 8: Returns statistics of the sample distribution and the reference original distribution, throughout the training of CeSoR over different benchmarks, for different values of $\beta \in [0.05, 0.3]$.

### D.3 Sample Efficiency

An important aspect of the CEM is its increase of sample efficiency (Section 3.2). While the results in Section 5 already demonstrate that CeSoR learns better and faster than the standard GCVaR, here we measure the effective sample size directly. While PG always uses the entire batch, and GCVaR always uses at most $\alpha$ of the episodes, Figure 9 shows that CeSoR manages to optimize $\text{CVaR}_\alpha$ while using more than $\alpha$ percent of the data. Note that even beyond the risk level scheduling (which ends after $\rho = 80\%$ of the training), the CEM still allows for more than $\alpha$ percent of each batch to be used.

Note that GCVaR effectively uses *less* than $\alpha$ episodes in a batch if multiple episodes $\{\tau_i\}$ satisfy $R(\tau_i) = q_\alpha$ – since the contribution of any such episode to the gradient in Equation (3) is $0$. In the extreme case, as discussed in in Section 3.1 and Appendix E.2, all the worst $\alpha$ episodes are identical, and the whole loss gradient is identically $0$.

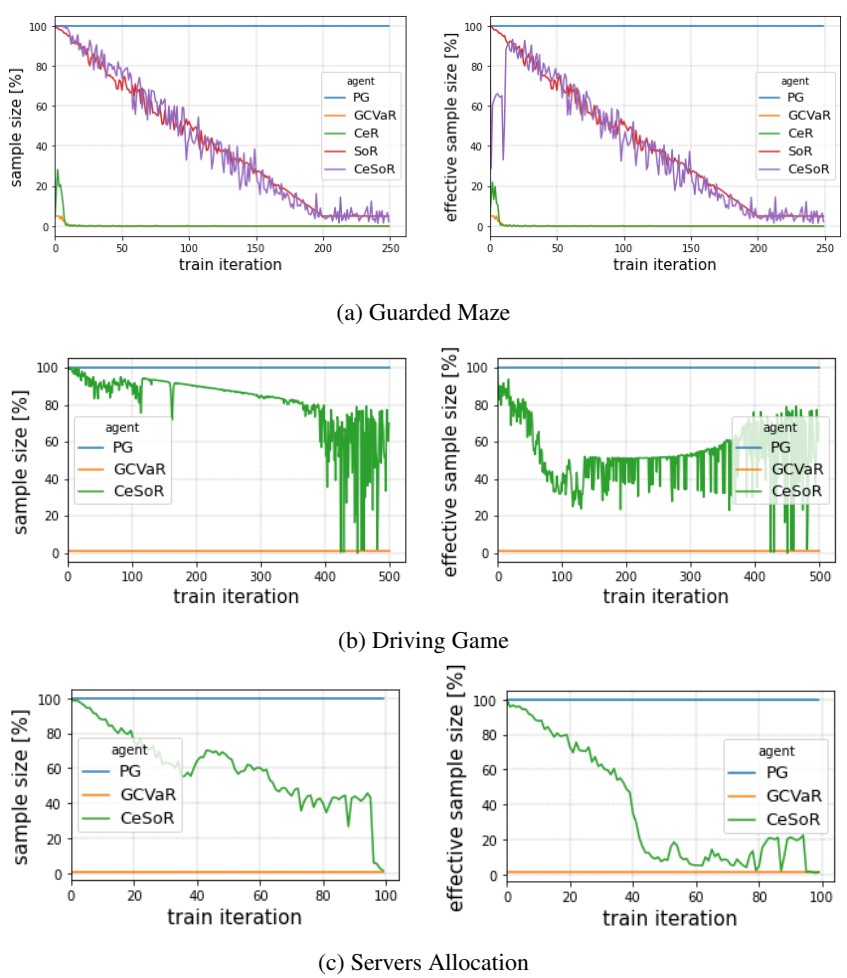

(a) Guarded Maze

(b) Driving Game

(c) Servers Allocation

Figure 9: Left – sample size: the percent of episode samples (out of $N = 400$ episodes per training iteration) used by the optimizer. Note that only returns $R(\tau_i) < q_\alpha$ are counted (strict inequality), since the contribution of episodes with $R(\tau_i) = q_\alpha$ to the loss is 0 (Equation (3)). Right – *effective* sample size: this takes into account the IS weights: the effective sample size equals the number of equally-weighted independent samples needed to obtain the same estimation variance [Kish, 1965, Leinster, 2014]: $n_{eff} = \left(\sum_i w_i\right)^2 / \sum_i w_i^2$. Note that for equal weights, $n_{eff} = n$.

## D.4 Risk Characterization

The CEM not only allows CeSoR to sample the most relevant environment conditions for CVaR optimization, but also allows us to characterize the conditions that correspond to the risk level $\alpha$. This enhances our understanding of the problem and may help us to anticipate poor returns in advance.

Figure 10 presents the evolution of the sample distribution parameters $\phi$ throughout the CeSoR training process in the various benchmarks. In the Guarded Maze, for example, $\phi$ goes back to its original values once the agent behavior converges, which teaches us that a risk-averse agent can be entirely insensitive to the environment conditions. In the Driving Game, on the other hand, the agent must still beware a leader that applies many turns and emergency brakes. Furthermore, the CEM provides the connection between the risk level of interest ($\alpha$) and the corresponding values of $\phi$ (e.g., how many turns and brakes it takes to bring us to this risk level).

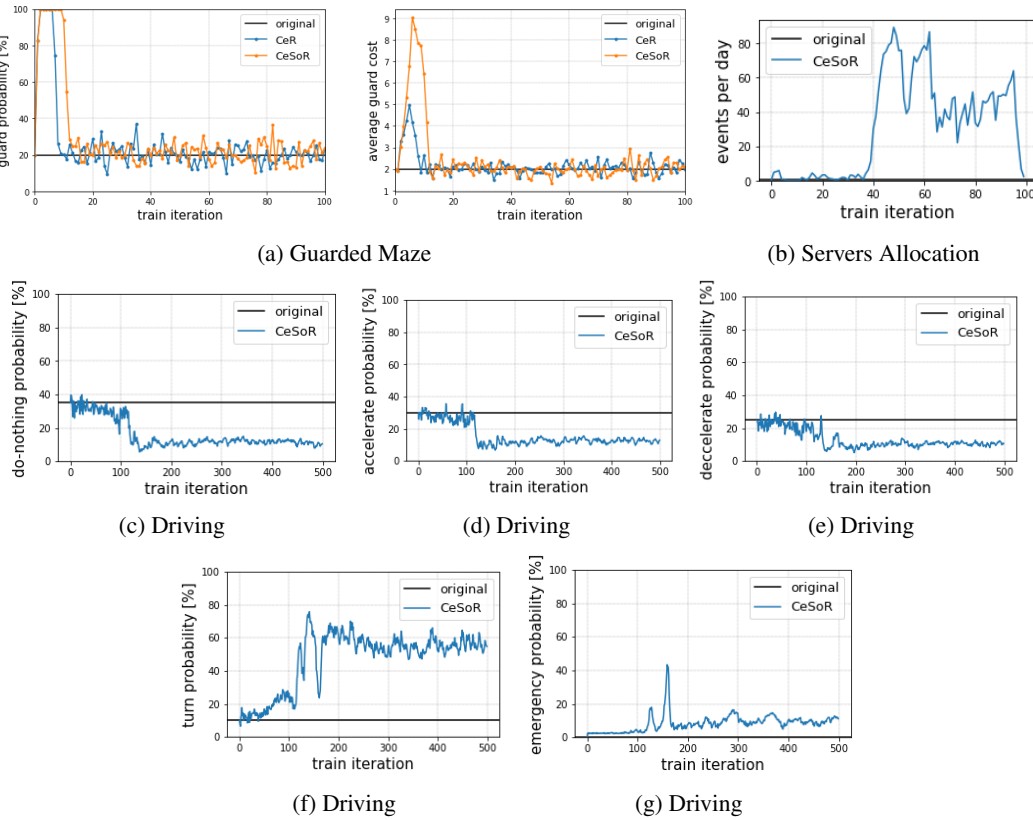

(a) Guarded Maze

(b) Servers Allocation

(c) Driving

(d) Driving

(e) Driving

(f) Driving

(g) Driving

Figure 10: The evolution of the CE distribution parameters $\phi'$ throughout the training in various benchmarks.

# E  The Guarded Maze: Extended Discussion

## E.1  Implementation Details

In this section we specify the implementation details of the Guarded Maze. The full code is available in the gym environment and the corresponding jupyter notebook.

**The Guarded Maze benchmark:**  The benchmark introduces a maze of size $8 \times 8$, with the walls marked in gray in Figure 1d. The target is a $1 \times 1$ square marked in green. Every episode, the initial agent location is drawn from a uniform distribution over the lower-left quarter of the maze. Every time step, the agent can walk in one of the directions left, right, up and down, with a step size of $1$, and an additive normally-distributed noise with standard deviation of $0.2$ in each dimension. That is,

$$s_{t+1} = s_t + a_t + (\epsilon_1, \epsilon_2)^\top$$

where $s_t, a_t \in \mathbb{R}^2$ and $\epsilon_i \sim \mathcal{N}(0, 0.2^2)$ ($i \in \{1, 2\}$). A step that ends in a wall is cancelled, and the agent remains in its place.

Every time-step, the agent observes its location $s_t$. In practice, we use a soft (continuous) one-hot encoding of the agent location in the maze, calculated as a 2D interpolation between the 4 nearest points of a $8 \times 8$ grid, represented as a corresponding $8 \times 8$ matrix. That is, if the agent is located between the grid points $(i, j), (i, j + 1), (i + 1, j), (i + 1, j + 1)$, then all the other elements of the matrix are set to $0$, and these 4 elements are assigned positive value that are summarized to 1, according to the relative location of the agent between them. Note that the locations of the target and the guarded zone are constant, and are not given as input.

An episode ends either when reaching the target or after 160 time-steps. The rewards are specified in Section 5.1. The return of an episode is the sum of its rewards (i.e., no discount factor). The maze is designed such that the $mean$-optimal strategy is taking the shortest path to the target, where the

expected cost of crossing the guarded zone is $E[C_1 C_2] = \phi_1 \phi_2 = 0.2 \cdot 32 = 6.4$ – smaller than the additional cost of the longer path. The $CVaR_{0.05}$-optimal strategy, however, is to take the longer path, since sometimes short cuts make long delays [Tolkien, 1954].

**Algorithms implementation:** The training algorithms are specified in Section 5. In the maze benchmark, all of them are applied to a linear model that takes as an input the one-hot encoding described above ($\in \mathbb{R}^{64}$), and is followed by a softmax operator with temperature $T$. That is, $P(a_j; \theta) = exp(T y_j) / \sum_{j'} exp(T y_{j'})$ (where $1 \leq j \leq 4$ and $y_j$ is the corresponding output of the linear model $F_\theta$). We set a constant $T = 1$ over the whole training, and $T = 0$ (i.e., choosing the max-probability action) for validation and test episodes.

The CE module in CeR and CeSoR controls the parameters $\phi$ of the Bernoulli and the Exponential distributions. Note that the module is aware of the original ("true") values of $\phi$, but not of their semantic meaning in the maze (e.g., it is not aware that high values are "bad", or that they only affect the agent through the guarded zone). The sample parameters update using the moments-method is as simple as $\phi \leftarrow (mean(C_1), mean(C_2))$, calculated over the episodes selected by the CE (Line 12 in Algorithm 1).

### E.2 Detailed Results

Figure 11 shows the distribution of the trained agent returns over the test episodes in the Guarded Maze (note that the left tail of this distribution is displayed in Figure 1a. Figure 12 shows the mean and CVaR of the training and validation scores throughout the training process. Below we elaborate on the training dynamics in general, and the blindness to success in particular.

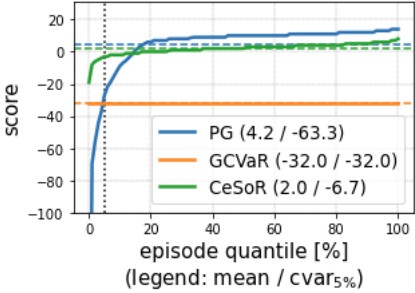

Figure 11: The full distribution of the trained agent returns over the test episodes in the Guarded Maze. Note that Figure 1a displays the left tail of the same distribution.

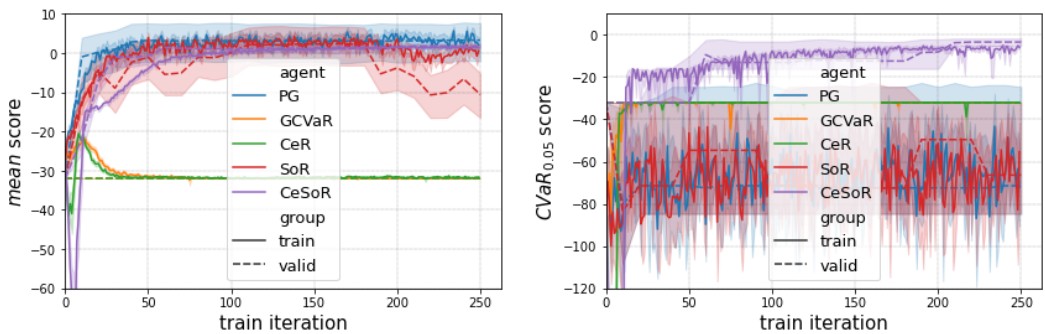

Figure 12: Mean and CVaR scores over the train and validation episodes throughout the Guarded Maze training. The shading corresponds to 95% confidence-intervals, based on bootstrapping over the episode-samples. Note that validation and train policies are not entirely identical, as the former deterministically chooses the action of max-probability (temperature $T = 0$), and the latter operates stochastically ($T = 1$).

**Blindness to success:** Section D.3 discusses the contribution of the *CE sampling* to the sample efficiency. Here we discuss the contribution of *soft risk level scheduling* to the sample efficiency, and in particular its prevention of *blindness to success*.

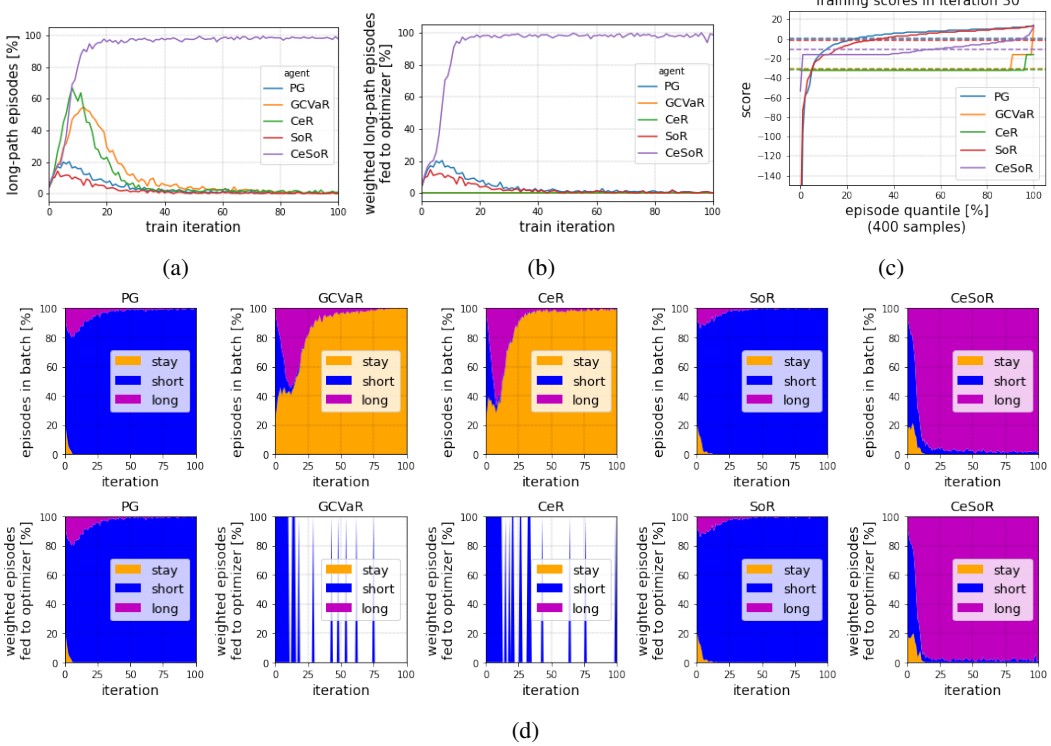

Figure 13: For the first 100 iterations of the Guarded Maze training, (a) the percent of episodes that reached the target through the long path; (b) the total weight of such long-path episodes that were fed to the optimizer (out of the total weight of episodes fed to the optimizer); (c) the returns distribution over the 30th training batch; and (d) percent of episodes (top) and total weight (bottom) for all 3 agent strategies (not only long path as in (a),(b)).

As displayed in Figure 13a, for all the agents in the beginning of the optimization process, around 10% of the episodes in every batch reach the target through the long path. At the same time, around 70% of the episodes reach the target through the short (and risky) path. As a risk-averse algorithm, GCVaR learns to avoid the short path, and the ratio of the long-path episodes increases accordingly – reaching up to 50% around the 15th batch (recall that in training episodes the actions are selected randomly according to the policy softmax output with temperature 1, which allows the agent to randomly reach the target). Nonetheless, as shown in Figure 13b, in *all* of the train iterations, *none* of the long-path episodes belong to the bottom $\alpha = 5\%$ episodes (which are fed to the optimizer), hence GCVaR never learns to prefer the long-path. This demonstrates the blindness of GCVaR to the successful long path.

In fact, after around 10 training iterations of GCVaR, all the bottom $\alpha = 5\%$ episodes in most batches already follow the stay-strategy (i.e., do not reach the target, nor take the guarded-zone risk), and achieve a constant return of $-32$ (Figure 13c). Note that according to Equation (3), this means that the loss gradient is identically 0. As shown in Figure 9a, the used sample size of GCVaR is indeed 0 after the 10th iteration, the effective sample efficiency is 0, and most of the changes in the agent from this point are attributed to the remaining Adam gradient momentum.

The soft risk level scheduling eliminates the blindness to success, and allows the optimizer to observe the long-path episodes (SoR in Figure 13b). However, at the same time, it reduces the risk-aversion of the agent, and the long path is no longer preferred over the short path. When the risk level reduces sufficiently, the agent may re-learn to avoid the short path, but the long path is no longer sampled at all and cannot be learned.

Only CeSoR manages both to observe the long-path episodes (thanks to soft risk level scheduling) *and* to prefer them over the short path (thanks to the risk-aversion induced by the CEM).

**Examples and visualization:** Figure 14 visualizes the policies learned by PG, GCVaR and CeSoR. While the policies are defined over all the continuous state space, the visualization is restricted to a discrete grid. Note that CeSoR and GCVaR behave similarly in the lower-left part of the maze, corresponding to guarded-zone avoidance; however, since GCVaR never observed the long path and learned its benefits, it fails to learn the CVaR-optimal strategy in the upper part of the maze.

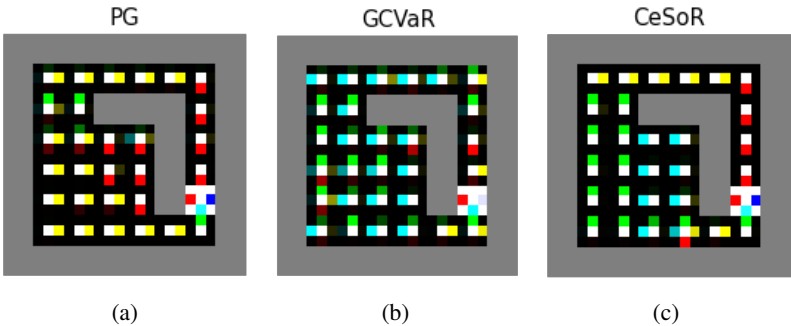

(a)                   (b)                   (c)

Figure 14: The policies learned by PG, GCVaR and CeSoR, visualized over a discrete grid within the continuous state space of the Guarded Maze. The colors brightness around each point in the grid corresponds to the probabilities assigned to the actions by the policy given this point.

Figure 15 shows a sample of test episodes for each of the trained agents. Due to the reduced risk-aversion of SoR (as discussed above), its best validation CVaR score was obtained early in the training, which may explain its non-smooth behavior in Figure 15.

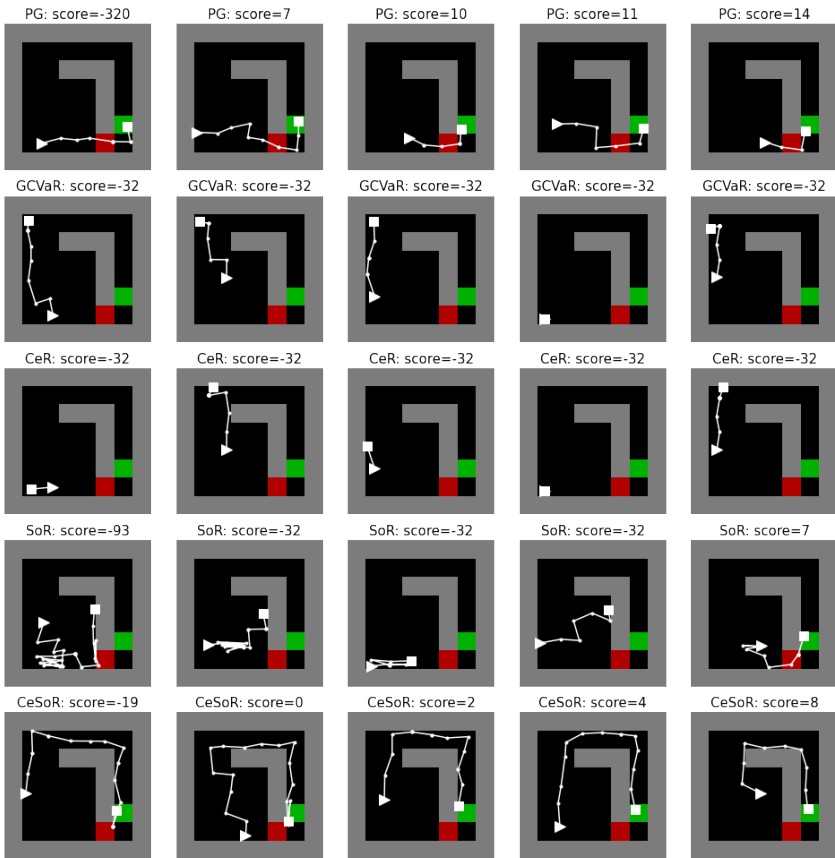

Figure 15: A sample of test episodes for each of the trained agents in the Guarded Maze.

# F    The Driving Game: Extended Discussion

## F.1    Implementation Details

In this section we specify the implementation details of the Driving Game. The full code is available in the gym environment and the corresponding jupyter notebook. Note that the leader behavior generation mechanism and the policy architecture are already specified in Section 5.

**Observation space**: the policy receives the following variables as inputs: relative position $dx, dy$, relative on-track velocity $dvx$, agent acceleration $ax$ and agent direction $\theta$.

**Action space**: the possible agent actions are (1) keep speed and steer; (2) accelerate; (3) decelerate; (4) steer left; (5) steer right. The acceleration and deceleration magnitudes $(+4m/s^2, -6m/s^2)$ were determined according to the typical acceleration value described in Singh et al. [2018].

**Rewards**: we use the rewards defined in Singh et al. [2018], with the parameters $r_1 = 0.5, r_2 = 0.05, r_3 = 0.1, r_4 = 0.5, r_5 = 1, r_6 = 0.5$. These parameters determine the scale of the 6 additive rewards of Singh et al. [2018], which correspond to staying behind the leader, staying close to the leader, keeping similar speed to the leader, keeping smooth agent acceleration, staying in the same lane as the leader, and staying on-road, respectively. We also add a new additive reward of size 5 for any time-step with overlap between the agent and leader cars, meant to penalize collisions – which are not explicitly expressed in the original rewards.

## F.2    Detailed Results

Figures 16-18 present a detailed analysis of the results of the Driving Game experiments.

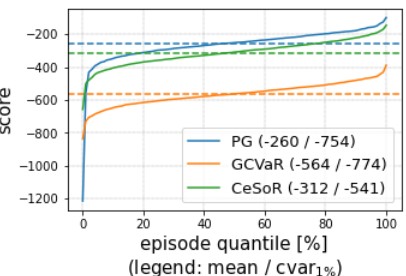

Figure 16: The full distribution of the trained agent returns over the test episodes in the Driving Game. Note that Figure 1b displays the left tail of the same distribution.

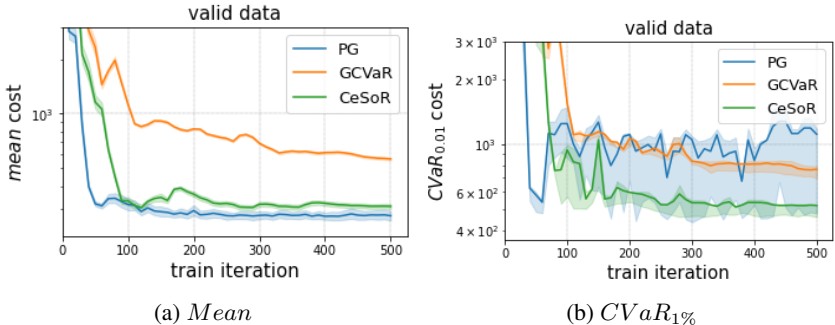

(a) $Mean$                                (b) $CVaR_{1\%}$

Figure 17: Mean and CVaR scores over the validation episodes throughout the Driving Game training. The shading corresponds to 95% confidence-intervals, based on bootstrapping over the episode-samples.

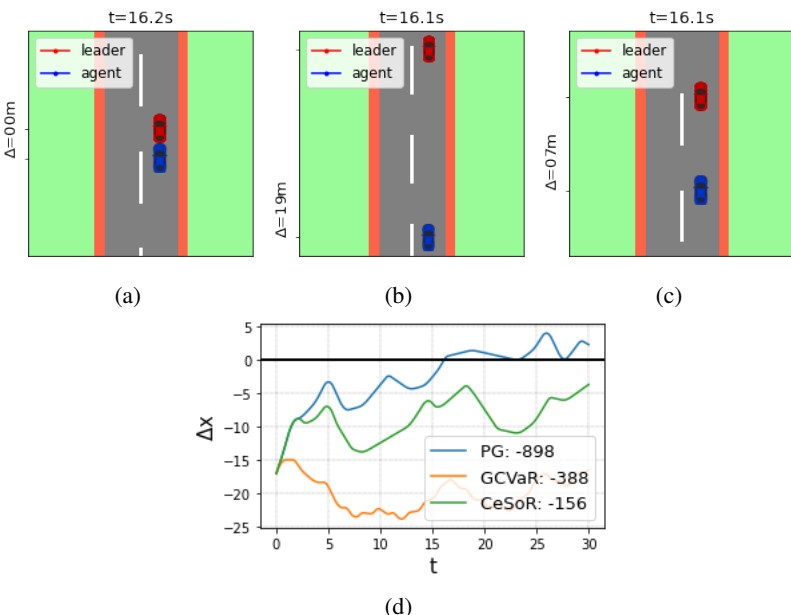

(a)        (b)        (c)

(d)

Figure 18: (a-c) A sample frame in a test episode in the Driving Game. All the agents deal with the same situation (the same sequence of leader actions, which happened to include a sequence of decelerations). While PG collides with the leader, CeSoR keeps a safe margin – without losing as much distance as GCVaR. Note that Figure 1e effectively displays these 3 frames together. (d) The agent-leader distance evolution in the whole episode, and the final episode score of each agent.

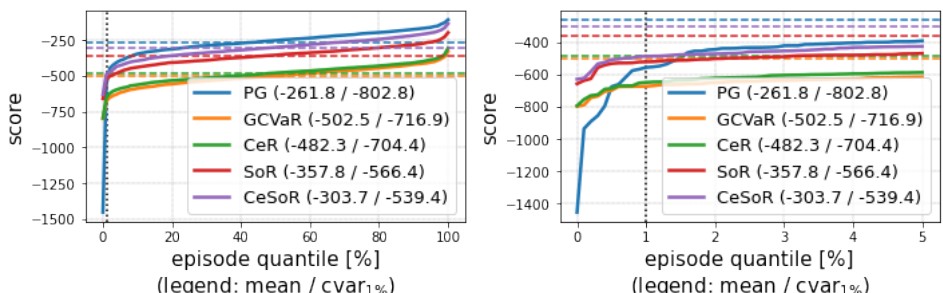

Figure 19: Additional ablation tests for the Driving Game: the full returns distributions (left) and zoom-in to their tails (right). Note that we reran the experiment for the ablation test, resulting in slightly different returns than Figure 1. Both CeR and SoR lose to CeSoR in terms of CVaR and mean, indicating the necessity of both soft risk and CE-sampler in CeSoR.

## G  The Computational Resource Allocation Problem: Extended Discussion

### G.1  Implementation Details

In this section we specify the implementation details of the Resource Allocation Problem presented in Section 5.3. The full code is available in the gym environment and the corresponding jupyter notebook.

The benchmark simulates one-hour episodes, where user-requests arrive randomly and the agent is responsible to allocate sufficiently many servers to handle them. Once a request is attended, its service time is distributed exponentially with an average of 1 second. Every second $t$, the number of arrivals is distributed $\sim Exp(\lambda_t)$, where the arrival rate $\lambda_t$ is itself an exponential moving average (EMA) of the (unknown) users interest $r_t$, with a typical decay of 5 minutes (i.e., $\lambda_t = \frac{299}{5 \cdot 60}\lambda_{t-1} + \frac{1}{5 \cdot 60}r_t$). $r_t = 3$ is usually constant, but an unpredictable event causes a peak load every second with probability $\phi_0 = \frac{1}{3 \cdot 24 \cdot 3600}$, i.e., every 3 days (or 72 episodes) on average. In case of a peak load we set the

momentary user interest to $r_t = 3 \cdot 300$, which means that the arrival rate doubles immediately to $\lambda_t = \frac{299}{300}\lambda_{t-1} + \frac{1}{300}r_t = \frac{299}{300}3 + \frac{1}{300}3 \cdot 300 \approx 6$, and then starts decreasing exponentially back to 3, with a typical decay of 5 minutes.

Every minute, the agent observes the number of active servers $3 \leq n^s \leq 10$ (initialized every episode to $n^s = 4$) and the number of pending user-requests in the system, and may choose to add or remove one server (or to keep the number of servers as before). Uploading a new server takes a 2-minute delay before the server is ready to handle requests. Removing a busy server takes effect once the server ends its current task. Note that the servers form an ordered list, and only the last server in the list can be directly removed. This constraint has little significance, since (1) the queue of pending requests is a global FIFO queue (i.e., the assignment only happens when a server becomes available – there is no separate queue per server); (2) the requests serving time is exponentially distributed, i.e., the remaining time of the current task is independent of the task history and thus is identical for all the busy servers at any point of time.

Denoting by $tts_i$ the Time-To-Service (TTS) latency of a request, the agent return is

$$R = -\text{user cost} - \text{servers cost} = - \sum_{i \in \text{requests}} tts_i - 2 \sum_{t=1}^{3600} n_t^s.$$

Once a request is assigned to a server, its serving time $\sim Exp(1)$ is independent of the agent decisions. Thus, to simplify computations and to reduce the noise, we measure the TTS of a request only as the waiting time between arrival and beginning of serving.

We set a target risk level of $\alpha = 0.01$, and train each agent for $n = 100$ steps. During the training, we gradually increase the episodes length $L$ from 15 to 60 seconds. The CEM controls the peak events frequency $\phi$, or equivalently, the number of peaks per episode (which is distributed $\sim Binom(\phi, L)$). The update function of $\phi$ is simply the (weighted) average number of peaks per selected episode, divided by the episode length. $\nu = 50\%$ of the episodes per batch are drawn from the original distribution $D_{\phi_0}$.

Note that at times of no peak-loads, the arrival rate is $\lambda = 3$ and the service rate equals the number of servers $n^s$ (since the service takes 1 second on average). Thus, in terms of queueing theory, any number of servers $n^s \geq 4$ guarantees that the expected number of requests in the system is $E[n^r] = 3/(n^s - 3) \leq 3$. In particular, this means that the policy learned by PG (see Section 5.3) chooses the minimal number of servers $n^s = 4$ that can handle no-peak demand, and adds resources only when required.

The agent policy receives a 9-dimensional vector as an input. The first 8 elements correspond to a one-hot encoding of the current number of paid servers $3 \leq n^s \leq 10$ (including new servers that are not finished uploading yet). The last element corresponds to the current number of pending user requests in the queue, divided by $10r = 30$ (the average number of arriving requests in 10 seconds of no peak-load).

### G.2 Detailed Results

Figure 1c summarizes the test scores of the agents, where CeSoR presents a reduction of 44% and 17% in the CVaR cost in comparison to PG and GCVaR, respectively. In addition, its average cost is only 7% higher than PG, and 33% lower than GCVaR. That is, CeSoR significantly improves the CVaR return without as a large compromise to the mean as in GCVaR. CeSoR also outperforms GCVaR in episodes both with and without peak events, as shown in Figure 20b below. As demonstrated in Figure 1f and summarized in Figure 22, PG and CeSoR learned to allocate a default of 4 and 5 servers, respectively, and to react to peak loads as needed; whereas GCVaR simply allocates 8 servers at all times.

Note that the CE task – sampling the bottom $\alpha = 1\%$ – is particularly challenging in this problem, due to the combination of very rare peak events and limited expressiveness of the Binomial distributions family. In particular, this family cannot guarantee the existence of a peak in a simulated episode without simulating *multiple* peaks per episode (i.e., $P_{\phi^*}^{\pi_\theta} \neq P_{\phi_0, \alpha}^{\pi_\theta}$ in terms of Section 3.2). Yet, CeSoR is demonstrated robust to the poor parameterization selection of $D_\phi$, as it presents a reasonable sampling (see Appendix D.2) and improves the returns CVaR.

Figures 20-23 present a detailed analysis of the results.

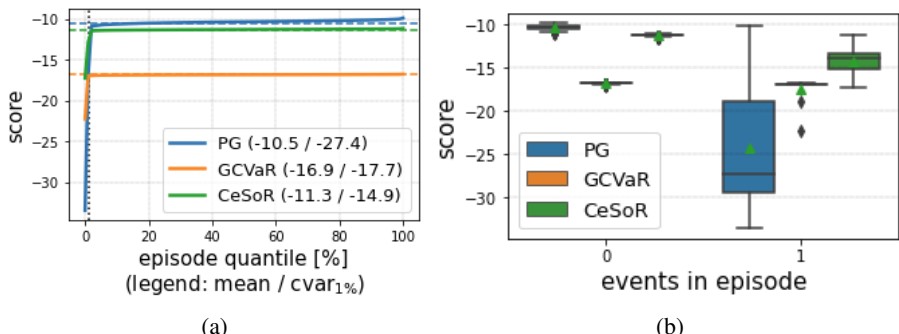

(a)                                                          (b)

Figure 20: (a) The full distribution of the trained agent returns over the test episodes in the Servers Allocation Problem. Note that Figure 1c displays the left tail of the same distribution. (b) A box-plot of the returns distribution for test episodes – separately for episodes with and without a peak-overloading event. CeSoR achieves the best scores in episodes with peak events.

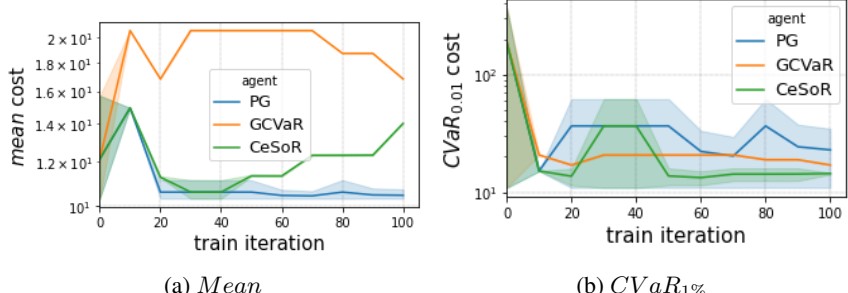

(a) $Mean$                                         (b) $CVaR_{1\%}$

Figure 21: Mean and CVaR scores over the validation episodes throughout the Servers Allocation Problem training. The shading corresponds to 95% confidence-intervals, based on bootstrapping over the episode-samples.

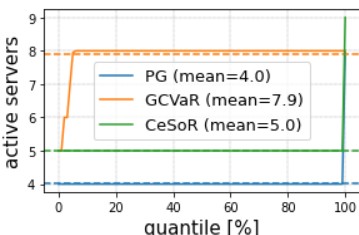

Figure 22: The distribution of the number of servers allocated by each agent, over all the time-steps in all the test episodes. GCVaR allocates 8 servers in advance, whereas PG and CeSoR typically allocate 4 and 5 servers, respectively, and add servers as needed in case of overloading.

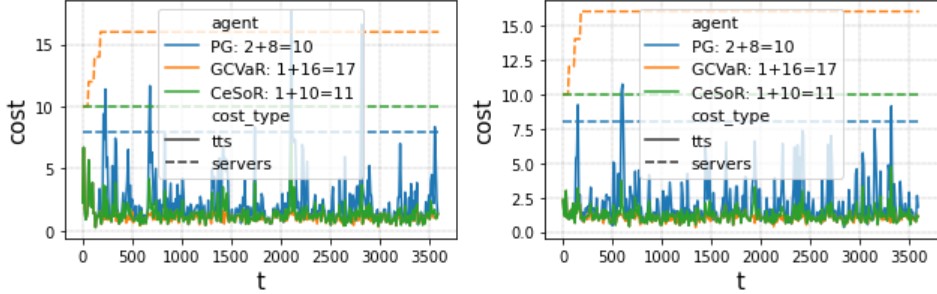

(a) Two episodes with no peak events: all agents ave near-zero TTS-cost, and servers cost corresponding to their policy (which is itself shown in Figure 22).

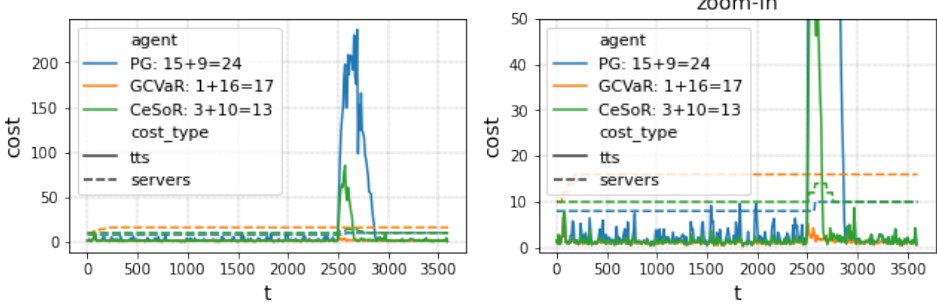

(b) An episode with a peak event (right: zoom in around the event). This figure presents the same episode displayed in Figure 1f, but normalizes the TTS and the servers allocation to the same units of cost, as defined by the benchmark. Notice that both PG and CeSoR react to the event with allocation of additional servers.

Figure 23: A sample of test episodes in the Servers Allocation Problem. The legends specify the TTS-cost, the servers-cost and the total cost.

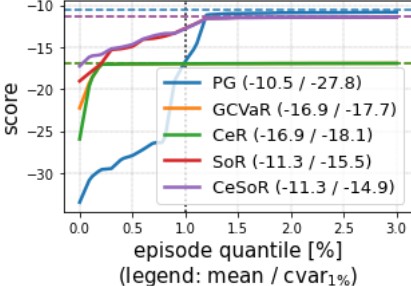

Figure 24: Additional ablation tests for the Servers Allocation Problem. Note that we reran the experiment for the ablation test, resulting in slightly different returns than Figure 1. Both CeR and SoR lose to CeSoR in terms of CVaR and mean, indicating the necessity of both soft risk and CE-sampler in CeSoR.

# H  Distributional Reinforcement Learning for CVaR Optimization

Many RL algorithms aim to learn the value $Q(s, a)$ of a state-action pair, representing the expected return from choosing action $a$ at state $s$. Then, given a state and a finite set of action candidates, the agent can choose the action with the highest value. In Distributional Reinforcement Learning (DRL), not only the expected return is learned, but rather the whole return distribution – conditioned on $s$, $a$ and the current policy. While standard DRL algorithms [Bellemare et al., 2017, Dabney et al., 2018b] still optimize the expected return and thus are risk-neutral, the learning of the whole return distribution encourages risk-averse variants as well [Dabney et al., 2018a].

A naive risk-averse DRL agent may simply use the learned return distribution to choose the action with the highest risk measure (e.g., CVaR) over the returns. However, notice that the return distribution is conditioned on the policy. Hence, similarly to other RL methods, the learned values become incorrect once we change the policy: the CVaR of the current action does not take into account the change in the next action. Thus, this naive approach would not truly optimize the CVaR.

Instead, a risk-averse DRL agent can train using a risk-averse actor, such that the learned distribution is consistent with the risk-averse policy. This approach is valid and was indeed used by Dabney et al. [2018a]. However, it suffers from similar limitations as CVaR-PG. Regarding sample-efficiency, CVaR-DRL considers only the bottom quantiles of the distribution, whose corresponding loss function assigns very low weights to all the returns except for the lowest ones, reducing the effective sample size. In particular, since there is no separation between low returns and high-risk environment conditions, still only a small portion of the data corresponds to high-risk, and it remains challenging to learn how to act under such conditions. Regarding blindness to success, CVaR-DRL is still prone to miss beneficial strategies: it still directs the actor policy according to the lowest returns rather than the hardest conditions, and learns the distribution with respect to that policy.

We implemented the methods mentioned above for the Guarded Maze benchmark, on top of the QR-DQN [Dabney et al., 2018b] implementation of Stable-Baselines [Raffin et al., 2021]. As shown in Table 1, none of the DRL variants improved the CVaR return even in comparison to the baseline CVaR-PG (GCVaR): the standard risk-neutral QR-DQN obtained similar returns to the risk-neutral PG; the naive DRL approach resulted in a noisy and seemingly-meaningless policy, obtaining worse returns than GCVaR; and the valid CVaR-DRL obtained identical returns to GCVaR.

These results support the discussion above, indicating that blindness to success and sample-inefficiency are general limitations in risk-averse RL, and in particular apply to DRL in addition to PG. We hope that our work will pave the way for other efficient risk-averse RL methods, beyond the scope of PG algorithms.

Table 1: A comparison of CeSoR test returns to both PG and Distributional RL methods, over the Guarded Maze benchmark. The first two methods are risk-neutral.

| Algorithm | Mean | $\text{CVaR}_{0.05}$ |
|---|---|---|
| PG | **4** | -63 |
| QR-DQN | **3** | -73 |
| GCVaR | -32 | -32 |
| CVaR-QR-DQN (only inference) | -32 | -39 |
| CVaR-QR-DQN (training+inference) | -32 | -32 |
| CeSoR | 2 | **-7** |