# OpenReview forum: "Efficient Risk-Averse Reinforcement Learning"
_NeurIPS.cc/2022/Conference — NeurIPS 2022 Accept_

### Official Review · Reviewer_YQ45 · 2022-07-11

**Rating:** 6
**Confidence:** 2
**Soundness:** 2 fair
**Presentation:** 2 fair
**Contribution:** 2 fair

**Summary:**

The authors study risk-averse reinforcement learning with the specific problem of how to keep risk-averse while using the collected samples efficiently. To this end, a soft risk-level scheduling mechanism is proposed where all samples are used at the initial stage, and it gradually shifts to high-risk (lower return quantile) samples. Additionally, a cross entropy method (CEM) is used to sample risky trajectories among soft risk samples. The proposed method is evaluated on three benchmarks to show its effectiveness.

**Questions:**

1. Some risk may caused by an insufficient exploration of the environment. What is the performance if using more advanced RL method, e.g. SAC instead of policy gradient?
2. How to understand the relationship of CEM and soft risk-level scheduling in Algorithm 1? It seems they are independent.

Update after the rebuttal.
It is clear now about the contributions and the relationship between CEM and soft risk-level scheduling.

**Limitations:**

No.

**Strengths And Weaknesses:**

**Strengths**

The risk-averse property plays an essential role in reinforcement learning (RL), especially for real-world problems. Therefore, the research topic of this paper is interesting and important for the RL community. The proposed mechanisms for improving risk sample usage are effective on the evaluated benchmarks. The authors also give a theoretical analysis of the ignorance of high-return samples, which leads to a local optimum.

**Weaknesses**

The writing need to be polished carefully. Although the paper is well-motivated, the methodology is hard to follow. Specifically, it is hard to understand how soft risk-level scheduling works together with CEM since the optimization in Algorithm 1 line 11:12 is independent of policy learning line 13:15. Additionally, a lot of details of the methodology, e.g. CEM are introduced in Problem Formulation sections, which further prevent the understanding. For the novelty, it seems unclear. The main contributions seem to be a heuristic risk quantile hyper-parameter scheduling and a sampling-based previous CEM.


Minor:

1. Symbol 1_{R(tau)<q_alpha} is undefined in Eq. 3. It seems to be the indicator function.
2. Grammar error: Are —> is in line 133.

---

> ### Author Response · Authors · 2022-08-01
> **Clarification of novelty**
>
> We thank the reviewer for their helpful comments. Please see our responses below, and in particular the clarification of our novelty. We will appreciate it if the reviewer can point to any unclarity in the paper regarding this issue, and reconsider their assessment of novelty in light of the discussion below.
>
> ### Novelty
> The reviewer wrote, “The main contributions seem to be a heuristic risk quantile hyper-parameter scheduling…”:
> * We point to a currently not discussed limitation of risk-sensitive optimization algorithms such as CVaR-PG - *blindness to success* - grounded by both theory (Section 4.1) and detailed experiments (Section 5.1). As stated by Reviewer QDcc, *“Theorem 1 is an important and interesting result that has standalone value for the general research area”*.
> * Based on this new understanding, we indeed suggest the simple and effective solution of soft-risk.
>
> The reviewer wrote, “...and a sampling-based previous CEM”:
> * This is not accurate: the standard CEM is not applicable to a non-stationary distribution (such as the returns distribution of a training agent). We had to develop a novel dynamic-target version of the CEM, as mentioned in Section 1 and described in Section 3. As mentioned by Reviewer WUYT, *“They extend the CE method for their particular setting”* (in fact, this extension can also be applied to other non-stationary problems).
> * In addition to the sampler’s novelty, the application of any sampler to focus on high-risk parts of the environment, as well as the theoretically-analyzed motivation (sample-efficiency), are important contributions by themselves. As stated by Reviewer QDcc, *“the application of CE and its corr. Proposition 1 is novel”*.
>
> Note that the conceptual novelties discussed above are also backed by the significant results of Figure 1 and Section 5.
>
> ### Responses to questions
> 1. **The PG reference**: As mentioned in the beginning of Section 5, the standard PG is not a competitor of CeSoR. Rather, it is used to demonstrate the inherent tradeoff between the different objectives (mean/CVaR). In all the benchmarks, PG performs well and achieves the best *mean* return (see Figures 1a-1c), using sensible policies (see Figures 11, 15, 19 in the appendix). In all the benchmarks, PG faces an *inherent risk* of the environment (the guard in the maze, the leader behavior in the driving, and the peak-loads in the servers), and compromises it in order to improve the *average*. In the only exploration-heavy benchmark - the Guarded Maze - PG learns the short-path policy (see Figure 11), which is mean-optimal, indicating it did not suffer from lack of exploration. The inferior CVaR of the risk-neutral PG comes from objectives difference, and does not reflect a failure to learn.
> 2. **CEM vs. soft-risk**: In addition to their independent motivations (sample-efficiency and blindness prevention, respectively), the soft-risk has a side effect of reducing the risk aversion in the beginning of the training, and the CEM reduces this side effect (as mentioned in the abstract, discussed in the the soft-risk paragraph of Section 3, and demonstrated in Section 5.1 and Figure 3). **We now extended the discussion in Section 3 to make this clearer**. Regarding Algorithm 1, the CEM changes phi, which changes the next-iteration trajectories, which determine the next PG step.
>
> ### Other comments
> * The separation between Sections 2 and 3 is of prior work vs. novel work. Hence, the original version of the CEM is introduced in Section 2, and our novel dynamic-target CEM is in Section 3. To prevent confusion, we now changed Section 2's title to "Preliminaries and Problem Formulation".
> * We corrected the grammar and defined the indicator notation. Thanks!

---

> > ### Comment · Reviewer_YQ45 · 2022-08-09
> > **Thanks for the response**
> >
> > I appreciate that the authors clarify the concerns, especially the relationship between CEM and soft-risk part.  They authors also rearrange and extend the main text which makes it much clear now. I have raised my score.

---

### Official Review · Reviewer_xSEv · 2022-07-18

**Rating:** 6
**Confidence:** 4
**Soundness:** 3 good
**Presentation:** 3 good
**Contribution:** 2 fair

**Summary:**

The paper studies policy gradient under the conditional value at risk (CVaR) objective. The proposed method contains two important components: (1) using CEM for the better sampling of data so as to obtain a better estimate of the value at risk; and (2) using soft-risk scheduling on alpha (the CVaR parameter) to address the blindness to success problem.


**Questions:**

- Figure 2 is a nice illustration. But could the authors provide how it is generated in detail? For example, how are R and C obtained for each method?
- Is there any additional reasoning for choosing alpha' the way presented in L13?
- This is just a small suggestion: For theorem 1, it might be more readable for the users to present it directly for B instead of negating the B. How should one interpret m_0? Does it depend on \alpha itself?
- For the experiments:
  - Why is GCVaR chosen as the PG method?
  - How big of a role does clipping play? Have the authors compared the performance of GCVaR without clipping with different methods?
  - In the first experiment, the authors have done an ablation study and compared CESoR with SoR and CEM separately. What about the second and third experiments? How does  CESoR compare with SoR and CEM?


**Limitations:**

Yes.

**Strengths And Weaknesses:**

Strength:

The paper proposed interesting heuristics to address the problem that CVaR is hard to optimize (especially in the RL setting). Algorithm 1 seems to be generic to any CVaR policy gradient method, which is ideal. The experiments have demonstrated the efficacy of the proposed methods.


Weakness:

The problem of estimating the VaR and the problem of zero gradients for optimizing CVaR is not new in the risk-sensitive learning field. For example: In "Adaptive Sampling for Stochastic Risk-Averse Learning (Curi et al.)", the authors have considered using adaptive sampling of the data to overcome these issues (both the estimation of VaR and the zero-gradient problem).  In "On the Convergence and Optimality of Policy Gradient for Markov Coherent Risk (Huang et al.)", the authors have showcased the zero-gradient problem of optimizing CVaR in a bandit setup (where the Markov CVaR is equivalent to CVaR). The proposed heuristics are interesting but it is unclear why certain design choices (e.g., the particular schedule of alpha') are made.

---

> ### Author Response · Authors · 2022-08-01
> **Responses**
>
> We thank the reviewer for their detailed comments.
>
> Also thanks for pointing out the two relevant works, which we now added to Section 1.1. However, please mind the significant differences between our works:
> * **Adaptive Sampling for Stochastic Risk-Averse Learning**: They indeed apply a sampling method to focus on risk, though not in an RL framework. In addition, **their discussion of zero gradients refers to a different phenomenon**: “only a fraction alpha of points will contain gradient information. The gradient of the remaining points gets truncated to zero” - this is the well-known phenomenon to which we referred as “sample inefficiency” or “throwing data away”. They do not address the phenomenon we call “blindness to success” - that is, zero gradients in the *remaining* alpha points, which completely eliminates the whole gradient and stops the entire learning.
> * **On the Convergence and Optimality of PG for Risk**: They mention the appearance of vanishing gradients in a specific example in the bandit setup, as part of their detailed discussion on non-unique stationary points. However, it seems that they do not analyze the phenomenon of the plateau in the loss; do not characterize it for general RL problems in terms of the returns distribution profile; do not make the separation between high-risk conditions and poor agent returns; and they leave the suboptimality gaps as a challenge for future research.
>
> ### Responses to questions
> * **Figure 2**: Every subfigure represents a batch of episodes; every point represents a single episode; C is the context of that episode and R is its return, both known to us during training (independently of which algorithm we use). Also note that Figure 2 specifically is only a qualitative illustration: as mentioned, it is analogous to the Maze benchmark, but does not present actual data. We now made this clearer in the caption.
> * **alpha’ scheduling**: The heuristic reasoning is a simple linear-decay from 1 to alpha (we briefly discuss that at the end of Section 4.1). The decay stops a while before the training’s end, so that the last part of the training has a stable objective. We did not try any other scheduling schemes (as mentioned in Section 4.1, this is left for future work). Our main take-away is that once we understand the problem of blindness to success, the solution can be simple and does not have to be particularly accurate.
> * **Theorem 1 and m0**: The gradient vanishes if there's a beta-tail barrier for beta>alpha (i.e., if all the beta lowest returns are identical). This may happen at any time during the training. We denote such a time by m0, and claim that there will be no further learning after iteration m0. Hence, *m0 is the first iteration where we encounter a barrier*. Smaller alpha makes the barrier more probable, thus it may come earlier (i.e., potentially decreasing m0).
> * **The GCVaR baseline**: As mentioned in Sections 3 (last paragraph) and 4 (first paragraph), unlike other methods, GCVaR has certain convergence guarantees that propagate to CeSoR if GCVaR is used as a baseline. In addition, GCVaR's simplicity arguably makes the empirical comparison cleaner.
> * **Clipping**: If the reviewer refers to weights clipping, note that GCVaR (with the default sample distribution) does not require weights and thus nor weights clipping.
> * **Following the review, we ran ablation tests for the rest of the benchmarks and added a corresponding appendix**. In both benchmarks (driving game and servers allocation), both CeR and SoR still lose to CeSoR in terms of both CVaR and mean. CeR performs similarly to GCVaR (but at least converges faster). SoR is closer to CeSoR but still loses in both metrics, which may be attributed to CeSoR’s increased sample-efficiency.

---

### Official Review · Reviewer_WUYT · 2022-07-20

**Rating:** 6
**Confidence:** 5
**Soundness:** 3 good
**Presentation:** 3 good
**Contribution:** 2 fair

**Summary:**

The authors provide different contributions, all related to the CVaR policy gradient method:
1) They analyze the phoenomenon named "blindness to success", which affects the CVaR gradient optimization.
2) They propose soft risk scheduling as a heuristic way of circumventing the problem.
3) They extend the CE method for their particular setting, obtaining the CeSor algorithm, which consists in applying the two enhancements on top of the original GCVaR algorithm.
4) They analyse the theoretical advantages in employing the CE method in the exact case, showing that it allows to reduce the policy gradient variance, thus, to reduce the sample complexity of a variance reduced PG algorithm.
5) They compare the performance of the proposed method with the original version on some simple domains, moreover, they provide an ablation study in order to analyse the contribution of the different components of the algorithm.


**Questions:**

- Can the authors please discuss how sensitive the approach is to the hyper-parameter $\beta$?
- In particular, is the choice of $\beta$ important to guarantee convergence to the optimal solution?
- If the approach is sensitive to this hyper-parameter, how should it be chosen?

**Limitations:**

As highlighted by the authors too, the main limitation is constituted by the necessity of having the possibility of conditioning sampling w.r.t. contexts. This requires to have a high-level control over the environments, an hypothes which can hold true in simulation, but which is usually more difficult to enforce in real-world environments.

Another limitation is constituted by the assumptions of Proposition 1, which are almost never satisfied in practical applications of the algorithm, as highlighted by the authors too.

**Strengths And Weaknesses:**

# Strenghts
The has the following points of strength:
- The empirical analysis shows the advantages of employing both the proposed enhancements, and an ablation study clarifies that applying just one of them alone is not sufficient to reach the optimal policy.
- The problem of blindness to success is analysed in a formal and exhaustive way.
- The theoretical contribution about CEM is sound and it allows to clearly highlight the advantages of employing this sampling strategy.

# Weakesses
The article presents the following weaknesses:
- The soft risk approach, while intuitive and justified from an empirical viewpoint is not analyzed from a theoretical perspective, thus, it can be considered just as an heuristic to avoid the problem.
- The theoretical results provided for the CE method hold only in the exact case, i.e., when the quantile extimation has no error and the CE method allows to match exactly the desired distribution.
- The role of the hyper-parameter $\beta$ is to guarantee a minimun number of samples in the CE update. However, this can introduce a bias in the distribution found by CE. The sensitivity of the approach w.r.t. to this hyper-parameter is not discussed.

## Minor
- Arguably, from the point of view of exposition, the article may benefit from a gradual introduction of the two main innovations propose the application of CE method and the risk-scheduling, instead of directly introducing the whole algorithm.
- Section 5.1 comment the performance of CeR, however, they are only showed in the appendix.

---

> ### Author Response · Authors · 2022-08-01
> **The $\beta$ hyper-parameter**
>
> We thank the reviewer for their helpful comments.
>
> Also thanks for pointing out the sensitivity of Proposition 1 to the quantile accuracy assumption. While this was already discussed in Appendix B, we now added a discussion in Section 4.2 as well.
>
> ### The $\beta$ hyper-parameter
> * **Intuitively**, every CE-iteration we focus on the beta-tail of the previous iteration, until we reach the alpha-tail of the reference distribution. Hence, intuitively, we expect exponential convergence to the desired alpha-tail, and larger values of beta are expected to cause only a small delay. Furthermore, even if the sampler is biased and samples from a tail less extreme than alpha, this should still provide an improvement over a neutral sampler.
> * **CEM convergence**: while having certain convergence guarantees, the rate of the CEM’s convergence is tricky to theoretically analyze for the general case. The dynamic-target in our CEM version sets an additional challenge for such a theoretical analysis (even though the target determined by the policy typically changes more slowly than the CEM sampler). These challenges are orthogonal to beta.
> * **CeSoR convergence**: the convergence proof in Appendix C does not rely on the performance of the CEM (and in particular not on beta), as the expected bias of the gradient is bounded for *any* sample distribution (as long as it has the same support as the original distribution).
> * **Practically**, we had simply used beta=0.2, which gives a decent sample size, yet expected to bring us to the 0.01-tail of the reference distribution within a few iterations. We hadn’t had to make any tuning for this parameter. Furthermore, Figure 5 in Appendix D2 shows that as desired, the sample-mean follows the reference-CVaR quite closely (up to the exceptions discussed at the end of Appendix D2).
> * **Following the review, we ran sensitivity tests for all the environments, and added a corresponding discussion in the appendix**. In the maze and the driving game, all $\beta \in [0.05, 0.5]$ provided similar test results, and only the highest ($\beta=0.5$) caused any visible delay in training convergence. In the servers allocation problem, the sampling task is more challenging due to the combination of small alpha (0.01) and poor distribution parameterization (Binomial, as discussed in Appendix G); there, beta<0.3 still performs similarly to the original CeSoR, but higher values fail to sample the tail, and begin to deteriorate towards GCVaR performance. Note that even under such a unique combination of poor choices (Binomial parameterization and very high beta), the failure of the CEM is easy to notice (in Figure 5c, the sample-mean fails to deviate from the reference-mean), and thus is easy to fix.

---

### Official Review · Reviewer_QDcc · 2022-07-22

**Rating:** 6
**Confidence:** 4
**Soundness:** 3 good
**Presentation:** 3 good
**Contribution:** 3 good

**Summary:**

- This paper studies the problem of optimizing CVaR-alpha of returns using the policy gradient method.
- The paper identifies and tackles two deficits of existing PG approaches to CVaR optimization:
1. the "blindness to success" phenomenon, which clips returns above the alpha-quantile of the return distribution and causes the corresponding gradients to be uninformative w.r.t. high-return scenarios -- the approach suggested by the authors is to start with a higher-alpha (e.g. 1) to allow PG to learn the high return scenarios, then gradually decrease the risk tolerance to the desired level as training progresses
2. sample efficiency, which is addressed by a cross-entropy method that learns to weigh low-return experience more than high-return ones during training
- Finally, the paper provides theoretical and empirical results on three domains that argue in favour of these two strategies

**Questions:**

1. clearly, the authors assert some additional assumptions about the structure of the problem to allow the CE method to gravitate towards the high risk areas. in contrast, distributional RL does not make such assumptions, and it can reduce the model uncertainty through bootstraps (unlike PG which inherently has high variance). how does the proposed cross entropy approach differ from learning a distribution over return? what is the advantage and the drawback of making such assumptions compared to DRL?
2. the cross entropy and "alpha-annealing" approaches seems to be largely orthogonal to one another. is there a stronger connection between them w.r.t. bias-variance trade-offs?
3. in relation to the weakness point made above, does the existence of function approximation errors limit the effectiveness of the alpha reduction method?
4. In sect. 4.2., variance reduction is connected to sample efficiency by showing that reducing variance reduces one term in the general error bound [Xu et al., 2020] . however, viewed in the same lens, doesn't annealing alpha play a contradictory role by increasing the bias of the CVaR estimate J_\alpha, and hence increasing the first term in that bound?

**Limitations:**

- limitations are briefly discussed in the conclusion of the paper, but it would be nice to discuss them in more details
- while this work is theoretical/abstract in nature, there could be some societal impacts in areas where RL is starting to become more practical (e.g. financial trading, health-care) - the implications of this with regards to the modeling assumptions (e.g. contextual MDP) used in this work should probably be mentioned

**Strengths And Weaknesses:**

Strengths:
- the problems described pertaining to the optimization of CVaR seem important to address
- the application of CE and its corr. Proposition 1 is novel to my understanding, and Theorem 1 is an important and interesting result that has standalone value for the general research area
- the Algorithm choices are clearly motivated by and well connected to the theory  (Th. 1, Pr. 1, Lem. 2)

Weaknesses:
- a number of important algorithms discussed in the related work section are not compared against in the empirical evaluation: specifically, the paper by Keramati et al., 2020 appears to be similar in terms of the problem being tackled as well as their methodology
- while I think the CE approach is interesting and appreciate the authors' theoretical insight and strong motivation for tackling the problem, my biggest concern is that it is largely unclear to me whether DRL and related approaches are already solving (or could readily solve) the same problem, and the distinction w.r.t. current work is a bit blurred (see question 1 below as well). I am willing to revise my score if additional experiments/discussion in the paper disprove this claim.
- the experiments are fairly low dimensional and it is difficult to determine if the effects of functional approximation errors could indirectly solve one of the problems this paper is tackling, on more challenging domains.

Organization:
- I think it would be helpful to the reader to provide a stylized example demonstrating empirically how CVaR optimization naively leads to the problems claimed in the intro. this might make it clear to the reader that the benefits seen in figure 1 do indeed arise from tackling the problems as claimed.
- I would also suggest to consider moving the discussion of blindness of success before the algorithm presentation, and use it as principled explanation for why a larger alpha will be necessary.

---

> ### Author Response · Authors · 2022-08-01
> **Distributional RL does not bypass the issues of risk-averse PG**
>
> We thank the reviewer for their detailed and helpful comments.
>
> ### Distributional RL
> Note that DRL in general is not a risk-averse algorithm, but rather risk-neutral. While the learned distribution can be leveraged to prefer risk-averse actions (e.g., distributional-rl.org, chapter 10, page 318), this approach suffers from similar difficulties to those of PG, as discussed below.
>
> * **DRL with CVaR on inference**: Consider a standard DRL agent (i.e., trained to optimize the *mean*), that is set to choose actions on inference according to *CVaR*. The distribution is learned wrt the training policy. Hence, similarly to other methods, the values would be incorrect once we changed the policy: the CVaR of the current action does not take into account the change in the next action. Thus, this naive approach would not truly optimize the CVaR of the return.
> * **DRL that uses CVaR consistently on both training and inference**: This approach still suffers from similar issues to PG.
>     * Regarding sample-efficiency, while not completely ignoring most of the data, still only a small portion of the data corresponds to high-risk conditions, making it difficult to learn how to perform well under them. Over-sampling of risk could still improve the accuracy of the learned distribution’s tail.
>     * Regarding blindness to success, this method is still prone to miss beneficial strategies: it still directs the policy according to the worst performance rather than the hardest conditions, and learns the distribution wrt that policy.
> * **We added corresponding experiments**: We ran both approaches mentioned above on the Guarded-Maze benchmark. We used the framework of stable-baselines3-contrib / qrdqn, and inserted the CVaR by replacing the mean when aggregating over quantiles in *policies.py* and in *qrdqn.py*. The results fit the discussion above.
>     * Switching to CVaR after training: this resulted in a messy and seemingly meaningless policy, obtaining worse CVaR than GCVaR.
>     * Using CVaR for both training and inference: **identically to GCVaR, this learned to avoid both the short path and the long path and obtained a constant return of -32**. These results indeed indicate that it suffers from the same limitations as GCVaR.
>     * We hope that the new experiments also address the reviewer’s concern about the lack of comparative baselines besides GCVaR and PG.
>
> * **In summary, we argue that blindness to success and sample-inefficiency are general phenomena in risk-averse RL, and in particular apply in DRL in addition to PG**. In this sense, in addition to our direct contribution to risk-averse PG, we hope to pave the way for other efficient risk-averse RL methods (as mentioned in the last paragraph of the paper). **We added the corresponding discussion and results to the appendix**.
>
> ### Responses to the other questions
> 2. **CEM vs. soft-risk**: In addition to their independent motivations (sample-efficiency and blindness prevention, respectively), soft-risk has a side effect of reducing the risk aversion in the beginning of the training, and the CEM reduces this side effect (as mentioned in the abstract, discussed in the the soft-risk paragraph of Section 3, and demonstrated in Section 5.1 and Figure 3). **We now extended the discussion in Section 3 to make this clearer**. It is not clear that this is a case of bias-variance tradeoff, since the bias of the soft-risk is a tool and not a compromise (see (4) below).
> 3. **Function approximation errors**: Even in complicated environments, the rewards may still be sparse or discrete, hence blindness to success still applies. Consider Montezuma’s Revenge for example: even if the agent can reach rewarding states, without the soft-risk the optimizer would simply discard them. In that sense, the environment complexity only increases the importance of sample-efficiency and of acknowledging beneficial strategies using soft-risk.
> 4. **Soft-risk causes bias**: The modified alpha intentionally modifies the objective, and thus indeed creates a bias. This bias is not arbitrary: it is designed to bypass the potential loss plateau (Section 4.1) and guide the policy towards the more successful strategies. Since it bypasses a problem in the loss-landscape itself, its “error” wrt the true gradient is in fact a desired deviation. For this reason, as mentioned in the beginning of Section 4.2, Proposition 1 refers to the last phase of the training (which uses the "true" alpha). As discussed above, before that phase, the CEM has another critical role in preservation of risk aversion. **We now elaborated on this in Section 4.2**.

---

> > ### Comment · Reviewer_QDcc · 2022-08-08
> > **Thank you for the clarification**
> >
> > Thank you for addressing the major concerns I had about the design of the work, and I have raised my score as a result (and possibly further depending on discussion).
> >
> > I agree with the findings of the additional experiments run to address my comments and those of the other reviewers. I think it will be a good idea to incorporate qrdqn and the approach of Keramati et al. as a baseline in all the experiments, especially since in my past investigations I have found DRL to work particularly poorly on the types of (near-degenerate) return distributions provided by grid-world domains. I would also recommend to evaluate the methods suggested by reviewer xSEv as well.
> >
> > I do like the overall approach taken and, while I do agree partly with the reviewer's concerns about the annealing trick for alpha being heuristic, the choice to bias with the soft risk is sufficiently supported by the theoretical claims. I think providing further discussion about the alpha' outside of the pseudocode would be helpful.
> >
> > That said, the main concept of the paper is quite nuanced and it can be difficult to follow without a clearer road map early on. I would still encourage the authors to provide some form of illustrations of the blindness to success phenomenon and explicitly how (and why) existing PG updates failed in the risk sensitive setting, e.g. through a stylized worked example or derivation of the PG gradient update where it is zero. This might help readers to navigate the paper (which reads quite dense) and understand the phenomenon a bit clearer early on.

---

> > > ### Author Response · Authors · 2022-08-08
> > > **Thanks for the detailed response**
> > >
> > > Thank you for the detailed response to our comments!
> > >
> > > * **QR-DQN**: We have indeed begun to investigate the phenomena and solutions of our work in the framework of DRL. While the same limitations indeed apply to this framework (as discussed in our previous response), DRL has different training dynamics and thus we intend to study the solutions carefully in a separate research. Furthermore, in DRL, we suspect that clever sampling may have benefits even without risk-aversion (e.g., in face of near-degenerate returns as you mentioned), thus we see this framework as promising for future work.
> > >     * BTW, is there any reason you referred to QR-DQN rather than IQN, besides being implemented in stable-baselines and used in our previous response?
> > > * **Further experiments**: following our extended experiments described in the first round of the rebuttal, we are now considering further experiments to extend our empirical evidence - both baselines and references as you mentioned - though we do not expect to have further results by the end of the discussion period tomorrow. Specifically regarding Keramati et al., do you happen to be familiar with an available code implementation? That could speed things up, but we did not find one.
> > > * **alpha’**: A dedicated discussion is a good idea. We’ll add it in Section 3 / Section 4.2 or in the appendix, according to space limitations. We’ll add a plot(x=iteration, y=alpha’) for a clear description of the scheme; and a discussion of why it actually bypasses the issue of Theorem 1, possibly through the Guarded Maze as a concrete example. We’ll also stress the point we mentioned in the rebuttal - that once we develop the understanding of Theorem 1, a simple and heuristic solution can indeed be sufficiently effective. Is there anything else you’d like to see in that discussion?
> > > * **Writing**: After addressing the questions regarding correctness and experiments, we now indeed consider both restructuring the paper (mostly switching the motivation of Section 4 with the solution of Section 3), and adding a guiding example as you suggested (probably using the Guarded Maze, unless you have another suggestion). Refactoring is always tricky, but we believe we’ll indeed be able to bring the paper to a clearer form without creating other sources of unclarity. Thanks!

---

> > > > ### Comment · Reviewer_QDcc · 2022-08-10
> > > > **Thanks for your response**
> > > >
> > > > Thanks again for your response, and apologies about the late reply.
> > > >
> > > > In terms of algorithm choice, I was only suggesting to extend the comparison from Guarded maze to traffic and server control domains, since I suspect (though could be wrong) that DRL should be less brittle on those problems than it would on the grid world domain. Given that Keramati's code has not been openly published, it would certainly be too much to ask for it during the rebuttal phase. However, I do recommend to a comparison in the camera ready if you can, because that algorithm seems like an obvious choice to address some of the problems also being tackled here as well, and could be seen as SOTA in some sense. My response to your claim "that once we develop the understanding of Theorem 1, a simple and heuristic solution can indeed be sufficiently effective" is that I do really like the connection you make between the theory and the annealing trick, which seems well-justified to me.

---

### Meta-Review · Area_Chair_mJMW · 2022-08-27

**Recommendation:** Accept
**Confidence:** Certain

**Metareview:**

Overall, the reviewers were satisfied with the author response and overall recommend acceptance.  However, there were many discussion points and nuanced details that arose during post-rebuttal author-reviewer discussion.  Reviewers would like to see these discussion points, clarifications, and requests for revision addressed in the camera-ready.  To this last point, I specifically highlight the writing/illustrative example discussion that the authors had with reviewer QDcc.  I fully agree that refactoring a paper is challenging, but ultimately, the suggested modifications will improve the accessibility of the ideas and contributions in the paper.

**Award:**

No

---

### Decision · Program_Chairs · 2022-09-14

Accept